# Metataxonomic Identification of Microorganisms during the Coffee Fermentation Process in Colombian Farms (Cesar Department)

**DOI:** 10.3390/foods13060839

**Published:** 2024-03-09

**Authors:** Carmenza E. Góngora, Laura Holguín-Sterling, Bertilda Pedraza-Claros, Rosangela Pérez-Salinas, Aristofeles Ortiz, Lucio Navarro-Escalante

**Affiliations:** 1Department of Entomology, National Coffee Research Center, Cenicafé, Manizales 170009, Colombia; lucio.navarro@austin.utexas.edu; 2Department of Post Harvesting, National Coffee Research Center, Cenicafé, Manizales 170009, Colombia; lhsterling22@gmail.com (L.H.-S.); ber.pedraza@mail.udes.edu.co (B.P.-C.); 3Cienciaudes Research Group, Faculty of Health Science, Universidad de Santander, Valledupar 200004, Colombia; ros.perez@mail.udes.edu.co; 4Department of Plant Physiology, National Coffee Research Center, Cenicafé, Manizales 170009, Colombia; aristofeles.ortiz@cafedecolombia.com

**Keywords:** washed coffee, microbiome, physicochemical analysis, attributes, quality, acidity

## Abstract

The metataxonomic diversity and microbial composition of microorganisms during the coffee fermentation process as well as their relationship with coffee quality were determined across 20 farms in the department of Cesar, Colombia, by sampling coffee fruits from *Coffea arabica*; Var. Castillo General^®^, Var. Colombia, and Var. Cenicafé 1. In each farm, the fruits were processed and the fermentation process took place between 10 and 42 h following this. Three samples of mucilage and washed coffee seeds were collected per farm during the fermentation process. The microorganisms present in the mucilage were identified using metataxonomic methods by amplifying the 16S rRNA gene for bacteria and ITS for fungi. The microorganisms’ morphotypes were isolated and identified. The analysis of bacteria allowed for the identification of the following genera: *Gluconobacter*, *Leuconostoc*, *Acetobacter*, *Frateuria*, *Pantoea*, *Pseudomonas*, *Tatumella,* and *Weisella*, as well as unclassified enterobacteria; the *Lactobacillacea* and *Secundilactobacillus* families were only identified in the Var. Cenicafé 1. For fungi, the top 11 genera and families found included *Hanseniaspora*, *Candida*, *Meyerozyma*, *Wickerhamomyces*, *Pichia*, *f-Saccharomycodaceae*, *f-Nectriciae*, unclassified fungi, and *Saccharomycetaceae*, which were only found in Cenicafé 1. A total of 92% of the coffee samples obtained scored between 80.1 and 84.9, indicating “Very Good” coffee (Specialty Coffee Association (SCA) scale). Farms with the longest fermentation times showed better coffee attributes related to acidity, fragrance, and aroma. During coffee fermentation, there is a central microbiome. The differences between the microorganisms’ genera could be influenced by the coffee variety, while the specific conditions of each farm (i.e., altitude and temperature) and its fermentation processes could determine the proportions of and interactions between the microbial groups that favor the sensory characteristics responsible for coffee cup quality.

## 1. Introduction

The coffee produced in Colombia is internationally recognized as a high quality, mild, washed arabica coffee due to the varieties cultivated, the climatic diversity in the cultivated areas, the production systems, and the harvesting and processing practices during post-harvest [1]. For this reason, since 2005, efforts have been made to establish an insignia, such as a Protected Designation of Origin, to guarantee consumers of the quality characteristics associated with the natural factors and traditional practices of the region in which the coffee is grown. Coffee consumers currently perceive the origin and type of processing as an additional attribute of the product that can directly influence purchase preferences [1], which is why it is essential to know the characteristics of the processes that influence the quality of coffee in certain places that are highly preferable by consumers [2].

The department of Cesar has an area of 22,905 km^2^ [3]; it is located in northeastern Colombia between 07°41′16′′ and 10°52′14′′ north latitude and 72°53′27′′ and 74°08′28′′ west longitude. There are two major mountain systems that are important for the country in terms of area and the ecosystem services they offer to the communities [3] where the coffee is grown: La Sierra Nevada de Santa Marta, which is located to the northwest of the department, and La Serranía del Perijá, which is a bordering mountain system between Colombia and Venezuela. The coffee growing process in Cesar plays a fundamental role in the agricultural and economic sectors due to the generation of both direct and indirect jobs. In this region, coffee growing is considered multi-ethnic, with the native populations Kankuamo, Kogui, Yupa, and Arhuaco being the main coffee growers [4].

The coffee from this area is characterized by a good sensory profile, high yield, and a Sierra Nevada denomination of origin with an organic seal [5]. For this reason, understanding the post-harvest processes, particularly fermentation, that the microorganisms are naturally involved in, the physical–chemical changes during the process, and the quality are important to generate added value and recognition of coffee growing in the region.

One of the most significant factors that affects the quality of the coffee drink in wet fermentation processing is the interactions that occur between pulped coffee and the microorganisms present during the fermentation process [6]. Various studies have been carried out worldwide to analyze microbial communities and their relation to different fermentation methods and their effects on the final quality of the coffee [7,8]. Coffee variety has also been considered [9,10], as well as fruit maturity and altitude [11,12], which are factors that affect microbial composition and, therefore, interfere with the biochemical changes associated with the metabolism of microorganisms.

The fermentation process plays a fundamental role in the degradation of the mucilage attached to coffee beans. The duration of the fermentation process in the farms will generally depend on the altitude and temperature of the farm [13]. Fermentation is performed by a diversity of native microorganisms present in the coffee fruit or by those who intervene in the process and are associated with the environment, water, air, machinery, containers, plant particles, or insects that are part of the diversity of microorganisms and produce significant changes due to the concentration of primary and secondary metabolites [6,14]. According to Puerta Quintero et al. [12], all the processes that occur in the fermentation stage are generated from the interaction of different factors; among the most relevant are the intrinsic quality of the coffee variety, the pH of the substrate, the microbial load, and the type of fermentation system. The quality of the coffee beverage is the result of both the intrinsic and extrinsic properties and factors from cultivation to post-harvest activities [15].

Additionally, in Colombia, recent interest in understanding the dynamics and diversity of microbial populations during coffee fermentation processes has resulted in the identification of predominant groups of bacteria and yeasts in most processes, as is the case for lactic and acetic acid bacteria (LAB-AAB) [16,17,18]. Due to their enzymatic potential, they degrade and/or ferment the substrates contained in the mucilage, favoring the production of primary, intermediate, and secondary metabolites, while consuming sugars, amino acids, lipids, citrate, lactate, pyruvate, and alcohol [19]. The enzymes involved, such as citrate permeases, citrate lyases, oxaloacetate-decarboxylases, pectin-lyases, polygalacturonases, and alcohol and aldehyde dehydrogenases, are those that allow for the formation of intermediate products in addition to the production of lactic, acetic, succinic, and malic acids, glycerol, butanediol, and acetoin [20], which provide differentiating properties to coffee quality. However, yeasts also play an important role in both the fermentation process and in the development of flavors and aromas in the final drinks [21,22,23] because they hydrolyze macromolecules, generating aroma precursors and reducing sugars, amino acids, and chlorogenic acids [24,25].

A previous study [10] characterized the microbiota of fermentation in three Colombian coffee varieties in Pueblo Bello, Cesar, showing differences in dominant microbial groups that influence the sensory characteristics of each variety. Other works have been carried out in the department of Nariño in the south of the country, in which De Oliverira Junqueira et al. [17], through molecular methods, identified the lactic acid bacterium *Leuconostoc* sp. as the most predominant during the entire fermentation process (0–48 h), and *Pichia nakasei* as the main yeast species during the process. To the north of the country, in a study by Cruz-O’Byrne et al. [16], *Leuconostoc* sp. was also found to be the main bacteria involved in the process, while the *Saccharomycetaceae* family was reported as the most abundant yeast, and *Kazachstania humilis* was identified for the first time in the coffee fermentation process. In the central zone of the country, Peñuela-Martínez et al. [18] also evaluated the fermentation in farms in the department of Quindío (Colombia), reporting 1349 bacterial genera, with enterobacteria being predominant at the beginning of fermentation; they mainly identified bacteria from the genus *Leuconostoc*, *Acetobacter*, *Tatumella*, and *Lactobacillus*. In fungi, genera of the *Saccharomycedaceae* family predominated.

The objective of this research was to characterize the microbiota present during the coffee fermentation processes in 20 farms in the department of Cesar with the usual post-harvest practices in each farm and to determine the relationships between the microorganisms, the quality control of the fermentation process, coffee variety, farm location, and coffee quality. For this, the microbial dynamics of bacteria, fungi, and yeasts were determined through culture-dependent and metataxonomic methods, and the physicochemical characteristics of the mucilage were also evaluated to finally correlate them with coffee quality and the physical and sensory characteristics of the beverage.

## 2. Materials and Methods

### 2.1. Location and Characterization of the Farms

A total of 20 farms (Fs) located in the municipalities of Agustín Codazzi, Pueblo Bello, La Paz, and La Jagua de Ibirico in the department of Cesar (Colombia) were visited. Sampling was performed in two coffee harvest periods: (1) between December 2021 and February 2022 and (2) between September and November 2022.

In each farm, the technical information of the crop and the traditional post-harvest practices carried out by the coffee grower were recorded. Information about the usual benefit process in each farm was obtained. The conditions of the fermentation process, as well as the material of the fermentation tank, the use of water during the process, and the times used for fermentation are recorded in Appendix A.

### 2.2. Sampling, Characterization of the Harvest, and the Fermentation Process

In all the farms, the manual collection of the coffee fruits from the plants was carried out in plastic containers or fique sacks that were deposited on the ground while the work was finished. In each farm, once the coffee growers harvested the coffee fruits, a sample of the coffee collected from the container was taken at random to determine the quality of the harvest using the Mediverdes^®^ tool (Agroinsumos del Café S. A, Bogotá, Colombia) and the Cromacafé^®^ (Agroinsumos del café S. A) color chart [26,27].

These coffee fruits were transferred to the coffee mill, weighed, and then entered the wooden hopper directly from the sacks; in no case were the varieties or collections made on different days mixed. The fruits were pulped and incorporated into the containers to be fermented either with or without water for the time that each coffee grower determined on a regular basis.

On farms that used water during wet processing, a sample of the water was collected in a 250 mL Nalgene (Thermo Scientific, Waltham, MA, USA) sterile bottle. These water samples were stored at 4 °C until processing for microbiological identification.

Under the coffee processing conditions that each farmer usually performs on his farm, the day’s batch was pulped. At the end of the process, 1.2 kg of pulped coffee was collected and processed by means of experimental mucilage removal, as previously described by Holguin et al. [10]. At this point, the first samples of mucilage (MT1) and seed (ST1) contained in the equipment corresponding to T1 (zero hours of fermentation) were taken. The remaining pulped coffee was deposited in the tanks, as the coffee farmer traditionally has to initiate the spontaneous fermentation process. Accordingly, the time determined by each farmer based on their experience was recorded. In the middle of the fermentation process, the second coffee sample was taken, which was processed in the mucilage removal equipment, and the mucilage (MT2) and seed (ST2) samples were obtained. At the end of the fermentation process, the intake of mucilage (MT3) and seed (ST3) was repeated, as previously indicated. These times varied from 10 to 42 h (see Appendix A).

Sampling was carried out under the guidelines of the Collection Framework Permit 01749 granted by the Environmental Licenses Authority-ANLA to the Universidad de Santander, Colombia.

In order to determine if a complete fermentation process would be carried out in terms of mucilage degradation, at the beginning of the fermentation process, the Fermaestro^®^ tool (Agroinsumos del café S. A, Bogotá, Colombia) [28] was used, which was introduced into the total mass contained in the fermentation tank and was left there until the final fermentation time was indicated by the farmer. These data were also recorded.

In all the farms, the seed samples (ST1, ST2, and ST3) were dried in a parabolic dryer [29] until reaching a humidity between 10 and 12%; then, they were stored in Ziploc bags and taken for quality analysis at the Quality Laboratory of the Cesar-Guajira Coffee Growers Committee. All mucilage samples (MT1, MT2, and MT3) were collected in triplicate in sterile Nalgene (Thermo Scientific Waltham, MA, USA) flasks and Falcon tubes that were stored at 4 °C in a portable refrigerator (Klimber 27 L) and transferred for preservation at −40 °C and 4 °C at the University of Santander in Valledupar (Cesar).

### 2.3. Physicochemical Analysis

For the mucilage samples taken at three different times (T1, T2, and T3), the following variables were determined in the fermentation tanks: temperature with a digital thermometer (Brixco, Berlin, Germany), pH with a portable pH meter HI98167 (Hanna Instruments, Woonsocket, RI, USA), and Brix degrees with an HI96801 digital refractometer (Hanna Instruments). For each measurement, three repetitions were made at different points in the fermentation tanks and the average and standard deviation were calculated. The Brix degrees were also evaluated in the field once the mucilage samples were taken. For each measurement, three repetitions were made and the average and standard deviation were calculated.

The acidity was determined in the mucilage samples MT1, MT2, and MT3 with the potentiometry technique using a 0.1 N alkaline sodium hydroxide (NaOH) solution with a 50 mL burette. Each measurement was carried out in triplicate and the average and standard deviation were calculated.

The total acidity value was expressed in mg L^−1^ equivalent to calcium carbonate (CaCO_3_) [30].

For the sugar analysis (sucrose, glucose, and fructose), the mucilage samples were centrifuged at 12,600× *g* RCF for 5 min and 1 mL of supernatant was taken and filtered through a 0.22 µm PVDF polyvinylidene fluoride membrane and kept at 4 °C until analysis. The samples were analyzed by means of high-performance liquid chromatography (HPLC) in the same way as reported by Holguin et al. [10]. Each measurement was carried out in triplicate and the average and standard deviation were calculated.

Physicochemical data from all 20 farms were grouped into three groups according to the final fermentation time. The first group contained the farms with up to 16 h of fermentation. The group 2 farms had 18 to 20 h of fermentation and the group 3 farms had a fermentation time of between 36 and 40 h. The data average and standard deviation were calculated in order to see the differences between the groups.

### 2.4. Metataxonomic Analysis

For the coffee mucilage samples (MT1, MT2, and MT3), DNA extraction was performed using 2 mL of sample. After, centrifugation was carried out at 16,000× *g* for 5 min. On the sediment, the extraction of genomic DNA (gDNA) was carried out with the QIAGEN DNeasy Powerlyzer Powersoil Kit. DNA quantification was performed by measuring the light absorption at 260 nm using NanoDrop™ equipment (2000-Thermo Scientific™). The gDNA samples were frozen at −20 °C.

High-throughput sequencing:

For the samples of DNA, amplicons were generated and they were used to characterize the microbiome taxonomic diversity. For bacteria characterization, the 16S rRNA gene molecular marker in the variable regions V3 and V4 was PCR amplificated using the following primers: Bakt_341F: CCTACGGGNGGCWGCAG and Bakt_805R: GACTACHVGGGTATCTAATCC [31]. Fungal diversity was determined via the amplification of the ITS molecular marker using the following primers: ITS3F (GCATCGATGAAGAACGCAGC) and ITS4R (TCCTCCGCTTATTGATATGC) [32].

The amplicons from the 16S rRNA gene and ITS end (PARE) libraries were obtained using the Herculase II Fusion DNA Polymerase Nextera XT Index Kit V2. For library construction, the sequencing library was prepared using the random fragmentation of the DNA, followed by 5′ and 3′ adapter ligation. Alternatively, “tagmentation” combines the fragmentation and ligation reactions into a single step that greatly increases the efficiency of the library preparation process. Adapter-ligated fragments are then PCR amplified and gel purified.

High-throughput sequencing was performed on the Illumina MiSeq platform, generating paired-end reads (PEs) of 300 bases each. The reads were cleaned up to a Q30 quality threshold and singletons and sequences less than 200 bases in length were removed using the Cutadapt version 3.5 program [33]. Sequences were analyzed with the Mothur program version 1.44, following the standard protocol for Illumina MiSeq libraries (SOP) [34]. PEs were assembled with the Mothur make.contigs tool and then aligned to the 16SSILVA reference database (Silva.nr v138) [35]. Subsequently, the VSEARCH algorithm [36] was used to remove the chimeric sequences. Sequences from nonbacterial lineages (sequences of mitochondrial, chloroplast, archaeal, and eukaryotic origin) were removed. The dist.seqs routine was used to group the reads into operational taxonomic units (OTUs), considering a distance limit between sequences of 0.03. The data were normalized with the normalize.shared command. The phylogenetic classification of the OTUs for fungi and bacteria with taxonomic assignment at the family and genus level for both bacteria and fungi was performed with the 16S Silva database (Silva.nr v138) [35], with a threshold of 80 (boot threshold 80) using the RDP Classifier Algorithm [37]. Coverage analysis was performed to determine sequencing coverage using Mothur and the rarefaction curve was also analyzed.

With the OTU-abundance data from each sample, beta diversity was analyzed using the Bray–Curtis distance and their ordination were visualized through nonmetric multidimensional scaling (NMDS). Differences in the community structure (beta diversity) between the coffee varieties were assessed with the permutational multivariate analysis of variance PERMANOVA [38]. Alpha diversity was examined with the indices Chao1, Shannon, Simpson, and observed species. Differences in alpha diversity were assessed using Kruskal–Wallis chi-squared statistics. Beta and alpha diversity analyses were performed in the Phyloseq and Microbiome packages of the RStudio program (version 2022.07.0) (https://posit.co/download/rstudio-desktop/, accessed on 10 April 2023) [39]. Statistical tests and graphs were carried out with the same program.

Taxonomic assignment at the phylum, family, and genus levels was performed for both bacteria and fungi.

### 2.5. Cultivation Counting and the Isolation of Microorganisms from Mucilage and Water

The MT1, MT2, and MT3 mucilage samples were processed to determine bacteria, yeasts, and mycelial fungi under the criteria of the current Colombian regulations [40,41]. Serial dilutions were made from the coffee mucilage samples, from 10^−1^ to 10^−4^, and 100 µL was inoculated on the culture media for each microbial group; mesophilic aerobes were cultured in nutrient agar (AN) and agar plate count (APC) (OXOID, Basingstoke, UK), and enterobacteria were cultured in MacConkey agar. Incubation was carried out at 28 °C for 48 h. After incubation, the CFUs (colony-forming units) were counted in a 10^−3^ dilution, counting each one of the CFUs that grew in the plates.

For the determination of lactic acid bacteria (LAB), first, an enrichment of the sample was carried out by taking 5 mL of mucilage and adding it to 25 mL of Man Rogosa Sharpe-MRS broth (MERCK, Darmstadt, Germany), then incubating it at 29 +/− 1 °C for 18 to 24 h at 80 rpm in a Shaker 1000 (Heidolph, Schwabach, Germany). Subsequently, serial dilutions were made up to 10^−5^, then 100 µL of each dilution was plated on the MRS agar medium and incubated aerobically and anaerobically at 28–30 °C for 48 h. After incubations, the CFUs were counted in the same way as described previously.

For the isolation of AAB, enrichment broths II and III were used (glucose, yeast extract, peptone, ethanol, pH 4.5; additionally, broth II contained acetic acid and had a pH of 3.5), adding (in equal proportion) broth and mucilage to the test tubes and incubating at 30 °C for 72 h at 80 rpm in a Shaker 1000 Heidolph [42]. Subsequently, a 100 µL aliquot was taken and inoculated on GEY-CaCO_3_-selective medium (2% glucose, 20% ethanol, 1% yeast extract, 3% CaCO_3_, agar) with the addition of 0.3% spiramycin 3 MUI (Labinco). Incubation at 30 °C for 10 days was followed by biochemical tests for identification [43]. After the 10 days, the CFUs were counted in the same way as previously described.

In order to grow and count the group of yeast and mycelial fungi Potato dextrose agar (PDA) (OXOID, Basingstoke, UK), yeast extract peptone dextrose agar (YEPD) (MERCK, Darmstadt, Germany) and Sabouraud 4% dextrose Agar (SDA) (MERCK, Darmstadt, Germany) were used. The incubation was carried out at a temperature of 28 °C for 96 h; after that, the CFUs were counted.

The microbial count data from all 20 farms were grouped into three groups according to the final fermentation time. The first group contained the farms that had up to 16 h of fermentation. The group 2 farms had 18 to 20 h of fermentation and the group 3 farms had fermentation between 36 and 40 h. The average and standard deviation were calculated in order to see the differences between the groups.

### 2.6. Identification of Microbial Groups from Coffee Mucilage

The identification of enterobacteria was carried out through the phenotypic criteria of colony-forming units (CFUs) and microscopic characteristics from the isolated bacterial strains. Additionally, biochemical tests were used, such as catalase, oxidase, OF-glucose, motility, and growth on MacConkey agar. As a complementary step, 1 CFU was taken, diluted in 1 mL of 0.85% saline solution, and inoculated into the API 20E gallery (BIOMERIEUX, Craponne, France). A dilution process of pure CFUs in 0.85% saline solution was carried out to identify spore-forming, Gram-positive bacilli. This was followed by another dilution on a 2 McFarland scale from which twice the quantity was extracted and added to the API50CHB/E vial. Subsequently, the gallery was inoculated. Furthermore, 12 tests from the API20E gallery were performed, as follows: β-galactosidase (ONPG), arginine dehydrolase (ADH), lysine decarboxylase (LCD), ornithine decarboxylase (ODC), citrate utilization (CIT), hydrogen sulfide production (H2S), urease activity (URE), tryptophan deaminase (TDD), indole production (IND), Voges–Proskauer test (VP), gelatinase production (GEL), and nitrite reduction (NO_2_).

Catalase, oxidase, and coagulase tests were carried out to identify spore-forming, Gram-positive bacilli.

For the identification of the LAB group after incubation, the total count was performed and each morphotype was characterized via macroscopic observation and microscopy. Biochemical discard tests were also performed, as follows: catalase, oxidase, coagulase, nitrate reduction, gelatin, mobility, and indole. Finally, the API technique using 50CH and API 50CHL (Biomerieux^®^), which contain 50 biochemical tests that assess assimilation, oxidation, and fermentation, was performed. Identifications were made with APIWEB^TM^ software.

For the AAB group after the incubation period, the characteristics of the morphotype were observed and a translucent halo formed around the CFUs. Smears and Gram staining were performed to confirm the morphology of the bacilli, as follows: medium and short Gram-negative, catalase tests, cytochrome oxidase, acetate oxidation, growth in 3% NaCl and D-mannitol broth, growth in glutamate agar, and growth in sugars such as L-arabinose, L-rhamnose, D-mannose, sucrose, and maltose [42,43]. Through these tests, the identification of the genera was achieved.

The identification of yeasts and mycelial fungi was carried out via the characterization of the isolated morphotypes by means of lactophenol blue staining and the observation of their microscopic characteristics. Then, the isolates were identified using the microbial DL-96II ID/AST system. Yeast colorimetry and turbidimetry were performed via the semi-quantitative analysis of antimicrobial MICs, following the manufacturer’s instructions (Zhuhai DL Biotech Co., Ltd., Zhuhai, China).

Mycelial morphotypes were identified from the development of colonies on PDA, SDA, and YEDP agar. The macroscopic recognition of the colonies (front and back), as well as their appearance (shape, size, color, and texture), was performed by reviewing the taxonomic keys [44].

Microscopic identification was carried out using the microculture technique via observing reproductive structures and using the taxonomic key of imperfect fungi [45].

Serial dilutions were carried out for the microbiological identification of the water samples. The most likely number (MPN) technique allows for the identification of total coliforms in Bright Green Broth according to the regulation. The number of mesophilic aerobes was also determined. The morphotypes were identified based on the macroscopic and microscopic characteristics and Gram staining [46,47]. Other bacteria were identified by plating in plate count media and eosin methylene blue or EMB agar using conventional biochemical tests that allowed for the isolation of different genera.

### 2.7. Physical and Sensory Analysis of Coffee and Coffee Cup Quality

Physical and sensory analyses were performed on the 60 seed samples (ST1, ST2, and ST3) from the 20 farms after they were dried and reached a humidity of between 10 and 12%.

The analysis of the physical quality of the green coffee beans was based on the determination of moisture, the percentage of loss, low quality, black and vinegar beans, beans infested with coffee borers, and the percentage of healthy beans according to what is established by the Colombian Institute of Technical Standards and Certification [48]. The sensory evaluation was carried out according to the Specialty Coffee Association (SCA) protocol with the participation of three Q-Grader-certified tasters by the CQI (Coffee Quality Institute), which belongs to the sensory panel of the Quality Laboratory of the Cesar-Guajira Coffee Growers Committee [49]. With this methodology, 10 coffee sensory attributes were recorded, as follows: fragrance/aroma, flavor, residual flavor, acidity, body, balance, uniformity, clean cup, sweetness, taster score, defects, and total, as well as sensorial, descriptors. The sensory quality expressed as the SCA total score was the response variable. The total score was the sum of each attribute score. The SCA total score and the coffee attributes data from all 20 farms were grouped into three groups according to the final fermentation time. The first group contained the farms that had up to 16 h of fermentation. The group 2 farms had 18 to 20 h of fermentation and the group 3 farms had fermentation between 36 and 40 h. The average and standard deviation were calculated in order to see the differences among the groups.

By using the Voyan Tools software [41] the coffee sensorial descriptors were analyzed in a word cloud.

## 3. Results

### 3.1. Characterization of the Harvest and Fermentation Processes

Appendix A shows the characteristics of the 20 sampled farms in the 4 municipalities of Cesar. Regarding the varieties of coffee planted on the farms, 55% corresponded to Var. General Castillo, 40% to Colombia, and one farm, F7, had Var. Cenicafé 1.

The farms were located between 1.116 and 1.938 m above sea level (m.a.s.l.). A total of 30% of the farms were located at an altitude between 1.116 and 1.390 m.a.s.l., 40% were at an altitude of 1.400–1.700 m.a.s.l., and the other 30% were between 1.740 and 1.938 m.a.s.l. A total of 50% of the farms evaluated did not use water for coffee processing, another 45% did use water during the process, and one farm (5%), farm F7, carried out submerged coffee fermentation, which consists of adding water to completely cover the pulped coffee mass. This measure was performed visually by the coffee farmer.

According to what was stated by the coffee growers, the processes in each farm are mainly established by the operational capacities of the mill and the coffee production on the farm; that is, the greater the amount of coffee collected, the shorter the fermentation times since they need to facilitate the spaces in the mill to process the next day’s collection, as was the case for farms F12 and F14, with 14 and 10 h of fermentation. In contrast, farms with a smaller amount of coffee collected prefer to leave their coffee fermenting for several days, as was observed in farms F6, F7, and F11, with fermentations between 36 and 42 h. Regarding the materials of the fermentation tanks, 50% of the tanks in which the fermentation took place were tiled, 35% were concrete tanks, and 15% were plastic tanks.

The characterization of the harvest with the Mediverdes^®^ tool in the 20 farms is presented in Appendix A. According to the classification of the tool, 75% of the farms had an excellent harvest and 20% had a good harvest. In 5% (F18), an average harvest was obtained.

There were some farms with incomplete fermentation according to the indicator Fermaestro^®^. In this case, most of the farms correspond to the shortest fermentation time. Farms F12, F14, and F15 had fermentation between 10 and 16 h and two farms had 18 h of final fermentation (F18 and F20).

### 3.2. Physicochemical Analysis

#### 3.2.1. Temperature in the Fermentation Tanks

The average temperature (Figure 1) in the fermentation tanks during the fermentation process ranged between 18.9 and 26 °C. At the end of the process, the highest temperature values were obtained in farms F11, F13, F1, and F9, ranging from 34 to 28.8 °C. A few hours after the start of the fermentation process, in some farms, a slight decrease in temperature was observed, mainly associated with environmental conditions, as these evaluation points were usually carried out at night. On average, 20.5 ± 1.93 °C was observed at 9 h of fermentation, 23 ± 3.12 °C at 18 h, and 26.3 ± 6.89 °C at 36 h of fermentation. Although the fermentation time varied depending on the farm, in general, an increase in temperature was observed in all fermentations between 1 and 10 °C. The farms where the temperature increased the most were farms F11, F17, and F1, with increments of 10.6, 8.4, and 6 °C, respectively. Furthermore, those that remained more stable were farms F2, F4, F7, and F8, with averages of 22.0 ± 0.01, 20.2 ± 0.01, 20.1 ± 1.1, and 20.8 ± 0.81, respectively.

#### 3.2.2. pH, Brix, and Total Acidity

The changes in pH, Brix, and total acidity during fermentation in the 20 evaluated farms can be seen in Appendix A.

Additionally, Figure 2 shows the results of the average pH and total acidity in the farms, grouped into three groups according to the final fermentation time.

For pH, the three fermentation groups showed a significant decrease between T1 and T3 at the end of the process. At the end of the process, independent of the fermentation time, the final pH was similar and around 3.8. In the farms with the longest fermentation time, this value was reached around 18 h, as well as in the other two groups of farms.

The Brix degree measurements show that in 80% of the samples (16 farms), a decrease in the dissolved solids and sugars was evidenced at the end of fermentation. On average, the decrease was clearly observed in the three groups and the behavior was very similar to that observed with the pH.

For total acidity, in the three groups, an increase was observed and the highest acidity value was reached in the group with the longest fermentation time, up to 42 h. At the beginning of the process, the values recorded were between 4.1 and 333.4 mg L^−1^ CaCO_3_. Particularly, the most notable increases in acidity occurred in farms F1 and F20, with values at the beginning of the process of 248.4 and 10.1 mg L^−1^ CaCO_3_ and ending the fermentation with 741.7 and 314.7 mg L^−1^ CaCO_3_, respectively.

#### 3.2.3. Sucrose and Reducing Sugars

The percentages of sucrose and reduced sugar in the mucilage are shown in Appendix A. Figure 3 shows the results of the average sugar data from all the 20 farms, grouped into the three groups according to the final fermentation time.

At the beginning of the fermentation process, the sucrose contents were between 0 and 0.75%, and, in 19 farms, the sucrose content decreased at the end of fermentation and the final total content was similar in the three groups. In F18, there was an increase from 0.04 to 0.23%.

The analysis results of reducing sugars composed of glucose and fructose monosaccharides in mucilage can be seen in Figure 3. The final total content was similar in the first and second farm groups, while the farms with longest fermentation time showed the lowest values and largest reduction. Particularly, in F18, an increase in reducing sugars was observed. On the other hand, the farms with the highest values of reducing sugars in mucilage both at the beginning and at the end of the fermentation process were F4 and F14.

### 3.3. Microbiologic Analysis

#### 3.3.1. Microorganisms Count

Appendix A shows the counts in terms of CFUs obtained for mesophiles, LAB, coliforms, yeasts, and filamentous fungi in the mucilage.

Table 1 shows the microorganisms count data, grouped into three groups according to the farm’s final fermentation time.

The LAB counts, in general, were higher than those obtained for the other groups, with an average of 6.29 log10 CFUs/mL, followed by yeasts with an average of 5.75 log10 CFUs/mL, mesophiles at 5.47 log10 CFUs/mL, coliforms at 3.92 log10 CFUs/mL, and, to a lesser extent, filamentous fungi at 2.8 log10 CFUs/mL.

In general, regardless of the group of microorganisms, there were more farms in which the counts of all groups of bacteria and fungi decreased as the fermentation time elapsed. In the fermentation time of up to 16 h, in the five groups of microorganisms, a decrease is observed between T1 and T3. In the group of up to 20 h of fermentation, only for mesophiles is no decrease apparent. Moreover, in the group of farms with fermentation times between 36 h and 42 h, there is an increase in microorganisms belonging to the mesophile and yeast groups.

From time 1 (T1) to time 3 (T3), a decrease in mesophile and yeast numbers was observed in 12 farms (60%); in the LAB group, a decrease was observed in 15 farms (75%). In three farms (15%), filamentous fungi were not observed in the counts (F7, F8, and F13) and of the remaining 17 farms, only one showed a slight increase in this group from T1 to T3 (F11).

Mesophiles in F6 and F9 were the predominant group, with averages in each farm of 6.19 and 5.94 log10 CFUs/mL, respectively. In both farms, the population increased from the beginning to the end of fermentation.

On the other hand, in yeasts, a decrease in the population was observed in 11 farms, and it was the predominant microbial population in F1, F2, and F3, respectively. In terms of filamentous fungi, these populations remained the same or decreased in 14 farms. Additionally, in F7, F8, and F13, they were not detected at any stage of fermentation.

Individually, farm F7 had the highest LAB count at T1 with 9.51 log10 CFUs/mL; for yeast and mesophiles, farm F6 at T3 showed the highest counts with 6.41 log10 CFUs/mL and 6.60 log10 CFUs/mL, respectively.

The development of coliforms and mycelial fungi in 55% of the farms decreased as fermentation time elapsed.

#### 3.3.2. Bacterial Metataxonomy Analysis

The analysis of sequencing coverage and rarefaction analysis with the 16S rRNA gene molecular marker presented percentages of greater than 96% (Appendix A), indicating that the sampling was sufficient and presented the appropriate characteristics to study the diversity in the samples. Likewise, it was possible to calculate the alpha diversity indices. Observed genus species or OTUs presented values between 172 and 1008, being higher in all farms at the beginning of fermentation. Farms F8 and F14 presented the highest values at the beginning of fermentation, with 1008 and 1002 OTUs, respectively, and at the end of the process, farms F10 and F17 had the highest OTUs with values of 436 and 457. No statistical differences for “Observed genus species” between coffee varieties (Kruskal–Wallis chi-squared = 1.5179; df = 2; *p*-value = 0.4681) were found.

The ACE richness estimator decreased from the beginning to the end of fermentation in the 15 farms, while an increase in wealth was observed on farms F8, F9, F10, F11, and F12. No statistical differences in the ACE index between coffee varieties (Kruskal–Wallis chi-squared = 1.2978; df = 2; *p*-value = 0.5226) were observed. For the Shannon diversity index, it was possible to show a decrease in all farms except F4; likewise, for the Simpson index, an increase was observed in farms F5, F15, and F16, while in the others, a decrease in diversity was identified from the beginning to the end of the process, particularly in F18. No statistical differences in the Shannon index between coffee varieties (Kruskal–Wallis chi-squared = 3.0791; df = 2; *p*-value = 0.2145) were observed. The Simpson index was the same at the beginning and at the end of the fermentation process, with a value of 0.90. No statistical differences in the Simpson index between coffee varieties (Kruskal–Wallis chi-squared = 1.8109, df = 2, *p*-value = 0.4044) were observed.

Beta diversity analysis was carried out using nonmetric multidimensional scaling (NMDS) (Figure 4), which allowed us to identify the relationship between the values obtained for alpha diversity and the coffee variety in each of the 20 farms evaluated. There is a significant difference in the genus microbial community structure (beta diversity) between coffee varieties (PERMANOVA *p*-value = 0.001). Few differences were observed, as shown by the dispersion of the points between the results of microbial diversity of the Var. Castillo general and Colombia, while a distant grouping was observed in the Var. Cenicafé 1.

The top 10 bacterial taxonomic assignments at the family and genus levels, with their relative abundances are shown in Figure 5 and Figure 6. The analysis at each farm was carried out at the initial, middle, and final points of the fermentation process, which allowed us to show the changes in the communities. At the family level (Figure 5), in all the farms, the bacteria of the *Lactobacillaceae* family were mainly identified; to a greater or lesser extent, the bacteria of the *Acetobacteraceae*, *Enterobacterales*, and *Erwiniaceae* families were also identified.

At the genus level (Figure 6), the metataxonomic analysis of bacteria allowed us to find the top 10 genera, where *Gluconobacter* and *Leuconostoc* stood out. Together, they were observed in a greater proportion in all the farms and evaluated times, followed by *Acetobacter*, *Tatumella*, and *Weisella*, and unclassified *enterobacteria* were present in most of the farms; however, the proportions were different in each farm. There is an assignment identified as “other”; in this, the genera from the *lactobacillaceae* family were present, as was observed in the 16S.trim.contigs.good.unique.good.fileter.uniqueprecluster.pick.nr_v132.wang.tax.summary.

In the wet processing of F1 and F11, a greater predominance of *Acetobacter* was observed in T3. On the other hand, in F2, *Leuconostoc* predominated during the process, while in F3, the bacteria of the genus *Weissella* predominated. *Secundilactobacillus* was predominant only in farm 7. In F4, bacteria of the genera *Pantoea* and *Leuconostoc* predominated. In F5, mainly the unclassified bacteria of the *Enterobacterales* and *Pantoea* genera were observed, and F6 had the highest predominance of *Tatumella* at T2 and T3; however, it could also be identified at T3 of F15, F16, and F18. In F7, with Var. Cenicafé 1, a decrease in the *Acetobacter* population was observed, while a progressive increase in the *Secundilactobacillus* population was observed. Farms F8, F9, F10, F12, F17, F19, and F20 showed an increase in the population of AAB *Gluconobacter* and *Frateuria* during fermentation. However, the abundance of the LAB *Leuconostoc* also increased during the fermentation process.

#### 3.3.3. Bacteria Isolates

Appendix A describes the mesophilic aerobic bacteria isolated at the three mucilage sampling points. In total, 40 different isolates corresponding to mesophilic aerobic bacteria were identified, after been found in different farms at different times (Figure 7). Gram-negative strains, such as *Stenotrophomonas maltophila*, stood out and were observed in 45% of the farms. In 40% of the farms, it was possible to isolate *Pantoea agglomerans*, followed by the presence of *Bacillus*, distributed as follows: *Bacillus firmus* in 25% and *B. subtilis* in 20% of the farms. The other bacteria of this group were only isolated into pure culture in three or fewer farms. Of the Gram-positive bacteria in 25% of the farms, *Micrococcus* spp. were observed in greater abundance in T1.

Table 2 describes the LAB; eight species were identified. A total of 100% of the farms exhibited LAB, either in the form of *Lactiplantibacillus plantarum*, *Leuconostoc mesenteroides*, or *Lactobacillus delbrueckii*. The farms that had *L. delbrueckii* did not have *L. plantarum*. The strains that were most frequently observed in the fermentation process of the farms were *L. plantarum*, followed by *L. mesenteroides*, *Levilactobacillus brevis*, and *Lactiplantibacillus pentosus* in the three sampling times. Additionally, the *L. plantarum* strain, in addition to being the most abundant on the farms (85% presence), was also observed during the vast majority of sampling times (67%). This, in turn, was one of the LAB that showed a decrease from T1 to T3. For the three remaining genera, 45% of the farms recorded the presence of *L. mesenteroides*. A total of 40% of the farms showed the presence of *L. brevis*. Likewise, 30% of the farms showed the development of *L. pentosus*. Other strains, with respect to their frequency on the farms, were lower, such as *Leuconostoc citreum*, which was present in 15%, and *L. delbrueckii* in only 10% of the farms.

Regarding the group of acetic acid bacteria or AAB, in 11 farms, this type of bacteria was not isolated (F1, F2, F3, F4, F5, F6, F8, F10, F12, F14, and F20) (Table 3). In total, four genera were identified, as follows: *Acetobacter* spp. and *Gluconobacter* spp., which were found in 25% of the farms. Another identified genus was *Saccharibacter*, which was found in two farms (10%) at both the beginning and end of fermentation, and *Acidomonas* was found in one farm (F9).

The results of the microbiological analyses of the water samples used for fermentation on the 10 farms showed that 15 bacterial genera were isolated; a total of 13 were Gram-negative, corresponding to enterobacteria and *Pseudomonas* spp., and two were Gram-positive: *Staphylococcus* spp. and *Micrococcus* spp.

The evaluation of the thermotolerant coliform in water using the most likely number technique indicated that F5 obtained the highest concentration of 460 MPNCF/100 mL, and for the mesophilic count, F10 registered a concentration of 4.72 log10/mL from AM. The main genera found were *Proteus mirabilis*, *P. vulgaris*, *Enterobacter* spp., and *Micrococcus* spp. for the first, and *Citrobacter freundii*, *C. gergoviae*, *C. murliniae*, and *Edwarsiella tarda* for the second.

#### 3.3.4. Yeast and Fungi Metataxonomy Analysis

The analysis of sequencing coverage with the ITS molecular marker for fungi presented percentages greater than 99% (Appendix A), indicating that the sampling was sufficient and included the appropriate characteristics to study the diversity in the samples. Likewise, it was possible to calculate the alpha diversity indices. Observed species or operational taxonomic units (OTUs) presented values between 61 and 629, unlike what was observed in the bacterial analysis. In the fungal communities, an increase in the species observed during the fermentation process was observed, with values from 285 to 319, 367 to 378, and 108 to 110 in farms F1, F3, and F11, respectively. In the other farms evaluated, a decrease in the species was evidenced from the initial point to the end of the fermentation process. No statistical differences for observed species between coffee varieties (Kruskal–Wallis chi-squared = 3.4776; df = 2; *p*-value = 0.1757) were found.

The ACE richness estimator varied between 105 and 2155, decreasing in 16 farms, whereas it increased in farms F1, F4, F5, and F7. No statistical differences in the ACE index between coffee varieties (Kruskal–Wallis chi-squared = 3.3426; df = 2; *p*-value = 0.188) were observed.

The results of the beta diversity analysis were carried out using nonmetric multidimensional scaling (NMDS) (Figure 8), which allowed us to identify a trend similar to that observed with the bacterial communities, since there were few differences between the diversity results of the genus coffee samples from farms with Var. Castillo general and Var. Colombia, while a distant grouping was observed in the samples of farms with Var. Cenicafé 1. There is a significant difference in the microbial community structure (beta diversity) between coffee varieties (PERMANOVA *p*-value = 0.001). These results indicate that there were marked differences in this group of microorganisms with respect to the other two varieties.

The top 10 metataxonomic assignments of fungi at the family and genus levels and their relative abundances are shown in Figure 9 and Figure 10. The analysis at each farm was carried out at the initial, middle, and final points of the fermentation process, which allowed for evidence of the changes in the fungal populations. At the family level (Figure 9), yeasts from the *Saccharomycodaceae* and *Phiciaceae* families were mainly identified; in F1, particularly at T1, unclassified fungi of the order Saccharomycetales predominated; however, at T1 and T2, the structure of the fungal community was of the fungi of the *Saccharomycodaceae* and *Phiciaceae* families, similar to what was observed in most of the processes. On the other hand, in the fermentation processes of F4, F5, F6, and F20, unclassified or “other” fungi were identified in the different stages, which may suggest the presence of new fungal species. At the genus level (Figure 10), a predominance was observed in the *Saccharomycodaceae* family in 19 of the 20 farms evaluated. In contrast, in F7, the presence of the *Saccharomycetaceae* and *Nectriaceae* families was evidenced.

#### 3.3.5. Yeasts and Filamentous Fungi Isolates

The fungi (yeasts and mycelial) were isolated and identified during the three fermentation times, which are listed in Appendix A.

Figure 11 shows the type of yeasts that were isolated in the natural fermentation process in the 20 farms.

A total of 100% of the farms showed *Hanseniaspora* and some *Candida* yeasts (*Candida* sp., *C. albicans*, *C. guilliermondii*, *C. kruesi*, and/or *C. tropicalis*). Finally, *Cryptococcus neoformans* was observed in 30% of the farms and the other yeasts appeared in less than three farms; among these yeasts are *Rhodotorula* sp., *Pichia*, and *Hanseniaspora*.

Appendix A shows the frequencies and times at which different mycelial fungi were observed in the farms: *Penicillium* sp. was found in 55%, *Rhizopus* spp., *Aspergillus fumigatus*, and *Aspergillus flavus* in 25%, and *Cladosporium* sp. in 20%. However, the highest number of *Penicillium* sp. strains was observed in the first and second fermentation times. A similar behavior was observed in *Rhizopus* spp. and *Aspergillus flavus*, where the presence of fungi was similar at times 1 and 2, but at the end of fermentation, they decreased.

*Cladosporium* spp. were present in 20% of the farms. Fungi, such as *Mucor* sp., *Trichoderma* sp., and *Fusarium* were seen in three or fewer farms. In three farms, F7 (36 h), F8, and F18 (18 h), no filamentous fungi were identified.

The mycelial fungi, although in low percentages, were present throughout the fermentation process of the mucilage at the three time points.

### 3.4. Quality Analysis

Appendix A shows the average humidity, total coffee cup SCA score, and sensory description for the 60 samples. The average humidity of the seed samples was 11.1%, so the samples were in the acceptable ranges of drying for the cup quality analysis.

The total SCA score data are grouped into three groups, according to the farm’s final fermentation time and are shown in Table 4. In this table, the data from farms F1 (T2), F1 (T3), and F 20 (T2), corresponding to defects associated with fermentation, mold, and contamination were not considered.

The average global score was 82.7. The three groups showed coffee considered “Very Good”, according to what is established by the Specialty Coffee Association [38], ranking between 82.3 and 83.7. A total of 3% of the samples achieved a score greater than 85, which is considered “Excellent Special Coffee”, 5% of the samples achieved values lower than 80 points, categorized as “Commercial Coffee”, and 92% of the samples achieved scores between 80.1 and 84.9, while the average highest scores (83.7) were observed in the farm group with the largest fermentation time (36–42 h) at the end of the fermentation. The best scores were observed in F2, F6, and F7.

In the total SCA score from all the sensory attributed evaluated factors, fragrance, flavor, and acidity, shown in Table 5, were the ones with the higher contribution to the total score. In this case, the highest cup acidity is observed in the samples with the longest fermentation time (T3), with a score of 7.73. These farms also were the ones that showed the highest total acidity in the mucilage physicochemical analysis (Figure 2).

With respect to the sensory attributes of fragrance and flavor, these are also related to the sensory descriptors. The coffee sensorial descriptors were analyzed in word cloud.

The Q-Grader certified tasters identified a particular sensory descriptor in the coffee cup samples from the farms with fermentation between 36 and 42 h at the end of the fermentation (T3). Figure 12 shows the word cloud that describes coffee sensory descriptors from those farms. The bigger the size of the word, the higher the frequency as a descriptor in the coffee cup of the farm. The sensory descriptors blackberries, fruity, citrus, and caramel are the most predominant. The others word clouds are not shown.

## 4. Discussion

The department of Cesar shows a large variability regarding the characteristics of the coffee farms, altitudes, temperature, fermentation times, and the material of the tanks where the wet fermentation takes place. However, the harvesting of the crop was appropriate based on the Mediverdes^®^ analysis. This could be a first indicator of a good cup quality of the coffee obtained.

The temperature characterization showed that the lower temperature variations seem to be an indicator of good fermentation. The farms F2 with Var. Castillo General, F6 with Var. Colombia, and F7 with Var. Cenicafé 1, and a fermentation time of 36 h achieved the best sensory quality results, and these showed the lowest temperature variation. This could indicate that a constant and stable temperature may favor the presence of certain groups of microorganisms that positively affect fermentation. The evidence may be in the metataxonomic analysis, in which the microbial populations of both the bacteria and fungi in these farms were stable at the three evaluation times. The increase in temperature during fermentation may be due to the number of microorganisms, as well as their type and growth rate, coupled with the high acidity due to the metabolites generated.

Because good quality results were obtained at the end of fermentation in farms with a constant temperature, the temperature in farm F1 (36 h fermentation) stands out, where the highest increase was observed throughout the long fermentation process and in which the defects (ferment and mold) associated with this occurred; it is possible that this high temperature favors the production of acidic metabolites and the growth of unwanted microorganisms, such as the *Citrobacter* and *Serratia plymuthica*-type enterobacteria isolated only on this farm. The genus *Serratia* has been isolated from water, soil, plants, and air and is known to secrete many enzymes, which are capable of inhibiting the growth of other microorganisms beneficial to the process since they degrade compounds in the substrate that are important for the development of certain microbial groups [50,51]. Therefore, it would be recommended to establish temperature control points to avoid the fermentation defects associated with the presence of unwanted bacteria. Nonetheless, additional assays under controlled conditions would be necessary to address the impact of temperature variations on quality features.

Regarding the Brix parameter, a decrease in this indicator is generated in response to the degradation of the organic acids, and the oxidation and fermentation of other substrates caused by the microorganisms. This also causes a decrease in pH that influences the various types of microorganisms present and their metabolites released during this stage, such as lactic, citric, succinic, malic, and acetic acids, ethyl acetate, acetaldehyde, glycerol, and ethanol, which can add value to the sensory characteristics of the final product [19,52]. The group of farms with the longest fermentation time, independent of the coffee variety, were also the ones with the final highest total acidity measured in the mucilage. They also had a high coffee cup SCA score at the end of the fermentation process. This score was related to the sensory attributes, as follows: acidity, fragrance, and aroma and special descriptors such as blackberries and fruity flavors.

Similar to temperature, the farms that showed lower variations in total acidity (F2 and F7) were the ones that also showed better characteristics of the drink, and the highest variations in this variable were associated with fermentation defects in farms F1 and F20. Notably, in these two farms, an accelerated growth of AAB was observed, which may explain the variations in this characteristic. Particularly in the farms with longer fermentation processes, it was possible to show stabilization in relation to pH after 18 h of fermentation, because at the end of the fermentation in the three groups of farms, the pH was very similar.

The farms with the highest values of reducing sugars in mucilage both at the beginning and at the end of the fermentation process were F4 and F14, and they also showed incomplete fermentation, as indicated by Fermaestro^®^. The degradation of these carbohydrates may be associated with the mechanisms used by microorganisms for mucilage degradation. In general, with respect to the sucrose consumption, small variations were observed in all the farms. However, in respect to reducing sugar in the farm group with fermentation between 36 and 42 h, the highest consumption and lower values are observed at the end of the process; this can be associate to the long fermentation time.

The number of microorganisms of the different groups isolated on the farms showed a greater presence of LAB, which coincides with the results obtained by Pereira et al. [46]. Yeasts, mesophiles, AAB, and, to a lesser extent, fungi were found, independent of the fermentation time. In contrast, Peñuela-Martínez et al. [18] also evaluated fermentation in farms in the department of Quindío, Colombia, and reported enterobacteria as the predominant group at the beginning of fermentation, and their abundance decreased over the course of the process. This indicates that the abundance of these microorganisms depends on the geographical areas. Peñuela-Martínez et al. [13] reported similar mesophilic aerobic bacteria and LAB counts with averages of 8 log CFUs/mL and yeast counts of 7.7, which are values higher than those found in our study.

In general, regardless of the group of microorganisms present, there were more farms in which a tendency was observed for the counts of all groups of bacteria and fungi to decrease as the fermentation progressed. However, the largest decrease was observed in the LAB group in the farms with fermentation between 36 and 40 h. In this group, a slight increase in mesophiles and yeast was observed. In the mycelial fungi, there was very little variation in the counts. When comparing the fermentation processes with longer times (36 and 42 h) to the fermentation processes with shorter times (10, 14, and 18 h), it could be expected that the longer the time, the higher the counts, but this was not the case. In general, this microbial count seems to be independent of the fermentation time. No major changes in the counts were identified between the various groups evaluated on each farm, and these observations are similar to what was reported by Peñuela-Martínez et al. [13].

The metataxonomic analysis of bacteria allowed us to find the top 10 genera with greater abundance, among which *Gluconobacter* and *Leuconostoc* (LAB) stood out. These were observed together in the greatest proportion in all the farms and times evaluated, followed by *Acetobacter*, *Tatumella*, and *Weisella*, and the unclassified enterobacteria present on most of the farms; however, the proportions were different on each farm. In the case of *Secundilactobacillus* sp., it should be noted that it was only found in the Var. Cenicafé. Additionally, this farm had one of the best sensory quality scores at the end of the process, with 84.0 SCA points.

In this study, *Lactiplantibacillus plantarum* was the species most frequently isolated in the fermentation process, together with *Leuconostoc*. Regarding the presence of *Leuconostoc*, it was also one of the most predominant bacteria during the entire fermentation (0–48 h) process in the departments of Nariño and Magdalena [16,17]. The LAB are recognized together with yeasts as the main transformers and contributors of aromatic compounds for coffee during the fermentation process by metabolizing sugars into acidic compounds, esters, ketones, aldehydes, and alcohols, among others; likewise, LAB contribute to the degradation of complex compounds, such as pectins found in high concentrations in coffee [20]. LAB groups have a metabolic activity that provides high concentrations of organic acids, such as succinic acid and lactic acid, in addition to ethanol and CO_2_. This activity positively influences lactic fermentation, along with hetero- and homofermentative processes in coffee; this type of fermentation is related to the delicate floral, fruity, and body perception of the drink [53].

The main genera and species identified within the Enterobacteriaceae family were *Pantoea agglomerans*, *Citrobacter freundii*, *Klebsiella pneumoniae*, and *Escherichia coli*, and similar results were obtained by Vilela et al. [54]. These correspond to ubiquitous strains in plant environments that can reach the fermentation process through external factors but have the ability to degrade the components of the mucilage through a mixed acid metabolic pathway. In the case of *Escherichia coli*, its presence is related to the lack of good manufacturing practices. On the other hand, the following stand out: *Stenotrophomonas maltophilia*, *Pasteurella pneumotropica*, *Hafnia alvei*, and *Chryseobacterium indolgenes*; their presence may be due to environmental factors and to the lack of application of good practices since they are found in plants, animals, and insects, among others [55]. The Gram-positive bacteria *Bacillus subtilis*, *Bacillus firmus*, *Bacillus cereus*, and *Bacillus megaterium* were identified, as previously reported in coffee fermentation by Viela et al. [48]. Regarding AAB, the genera *Acetobacter* spp. and *Gluconobacter* spp. were identified and the concentration of acetic acid generated during the coffee fermentation process may be due to the enzymatic degradation of the sugars and the alcohol released by yeasts and LAB. These compounds are used as nutrients by acetic bacteria that are strictly aerobic [56].

Regarding the bacteria identified in the processing water used by some farms, *Citrobacter* spp., *Enterobacter* spp., and *Klebsiella* spp. are enterobacteria that ferment sugars via a mixed acid pathway, which indicates that they have the ability to exploit the mucilage components through degradation and fermentation with the generation of different organic acids, alcohols, and gases; however, these genera are sensitive to highly acidic conditions, the secondary metabolites generated by LAB, and temperature changes. Among the most frequent species in water and mucilage found in this study are *Citrobacter* and *Enterobacter*, previously reported in [57,58].

In the metataxonomic analysis, the greatest abundance of yeasts found at the family level corresponds to *Saccharomycodaceae*, *Phiciaceae*, and *Saccharomycetaceae*, and genera such as *Hanseniaspora* sp., *Saccharomyces* sp., *Candida* sp., *Meyerozyma* sp., *Pichia* sp., and *Wickerhamomyces* sp., together with sp. and another group of unidentified yeasts, suggest the presence of new fungal species. This finding demonstrates the diversity in the existing microbiota in the coffee of the department of Cesar. Various investigations have shown the presence of yeasts such as *Candida parapsilosis*, *Saccharomyces cerevisiae*, *Pichia guilliermondii*, and *Hanseniaspora opuntiae* at the beginning and end of the fermentation process of both washed and unwashed coffee in Brazil [22]. In particular, in this work, 20 yeasts grouped into six genera were found, and the *Hanseniaspora* sp. was the most abundant yeast species in coffee fermentation in the department of Cesar. Its presence within the consortium of microorganisms could guarantee a better quality in the drink due to its high pectinolytic activity, excellent fermentative capacity, and the production of desirable metabolisms in the final product, such as acids, alcohols, pyridines, aldehydes, and furans, which results in a favorable tasting cup of coffee [59,60,61]. Cruz-O’Byrne et al. [16] also indicated abundant yeasts of the *Saccharomycetaceae* family such as *Kazachstania humilis* in coffee fermentation processes in the department of Magdalena. While Peñuela-Martínez et al. [18] identified the yeasts of the *Saccharomycedaceae* family as the most abundant, with 33 to 52%, on average, in the fermentation process.

The presence of *Hanseniaspora*, *Wickerhamomyces*, and *Pichia* in this study is similar to that reported in the communities of microorganisms found during coffee fermentation in China, Ecuador, and Brazil [62,63,64]. The genus *Pichia* has been reported as a dominant yeast in coffee fermentation in different countries and it is associated with the production of high-quality coffees with sweet, floral, and fruity characteristics [58,59,65,66]. Because it has pectinolytic activity, it helps in the degradation of the pectin present in the pulp and mucilage of the coffee by producing metabolites that diffuse into the interior of the coffee beans, favoring the formation of the flavor of the final drink. According to Kurtzman [67], *Meyerozyma*, also identified in this study, is a species of yeast capable of fermenting glucose, sucrose, and raffinose and of maintaining it at a mesophilic temperature (20–45 °C) during fermentation [68]. This genus was also detected in coffee fermentation processes in Australia and Brazil [22,69,70].

Regarding the mycelial fungi found and belonging to the Nectriaceae family (*Penicillium* sp., *Rhizopus* spp., *Aspergillus fumigatus*, *Aspergillus flavus*, *Cladosporium* sp., and *Fusarium*), these fungi have also been reported by other authors in fermentation processes in Brazil, where genera such as *Penicillium*, *Fusarium*, *Aspergillus*, and *Cladosporium* represented 42.6% of the genera found in the fermentation process [54,71]. It is important to be aware of the presence of species such as *Penicillum* and *Aspergillus* because, under some conditions, they are producers of Ochratoxin A-OTA [72].

The diversity analysis of the bacterial and fungal communities according to the coffee variety showed differences in the group of bacteria present in Var. Cenicafé 1, with respect to Var. Colombia and Var. Castillo. There was a relative abundance of bacteria of the genus *Secundilactobacillus* in the fermentation process, whereas *Acetobacter* spp. decreased; in particular, a high level of participation of bacteria classified as “Others” (corresponding to the *Lactobacillacea* family) was observed. Similar results were found in the group of predominant yeasts, with the *Saccharomycetaceae* family being the most important in the coffee fermentation process of Var. Cenicafé 1, which differs from the other two varieties. In Var. Castillo General and Colombia, the dynamics were more variable between times and the yeasts of the *Saccharomycodaceae* family predominated.

The differences between the farms were analyzed according to the fermentation time that is also related to the farm altitude, with higher altitudes resulting in longer fermentation times. The results indicated that the abundance of the microorganisms during fermentation is not related to the altitude or fermentation times, but depends on other factors. The farm with the shortest fermentation time at a low altitude and the farm with the longest fermentation time at a high altitude both showed the presence of similar microorganisms including fungi. In this case, to reach a complete fermentation, allowing the mucilage to be degraded completely, seems to be the important point.

The sensory analyses of the 60 coffee seed samples with the SCA methodology showed that “Very Good” coffees were obtained from the three varieties, a total of 92% of the samples obtained scores between 80.1 and 84.9. The average global score was 82.7%. The samples from Cenicafé 1 (36 h) stood out, showing a clearly different pattern of microorganisms, as observed in the beta diversity analysis and metataxonomic analysis and 84.0 SCA. On the other hand, the other two varieties Var. Castillo General ^®^ (F2—36 h) and Var. Colombia (F11—36 h), also show one of the highest scores between 83 and 84.25. The results also show that good practices such as a good quality of harvesting using the Mediverdes^®^ tools and the determination of the coffee’s final fermentation using the Fermaestro^®^ tool, allow us to achieve a very good cup.

In 50% of the farms, it was observed that the fermentation process allowed for an increase in the final sensory quality scores, and only in two farms (F1–F20) were the defects possibly associated with the increase in the population of bacteria of the genus *Acetobacter* and the interaction with other microorganisms during fermentation. In farm F1, at the fungal level, yeasts of the genus *Candida* also increased significantly, while in farm F20, unclassified genera were identified. It is possible that the interaction between acetic bacteria and certain fungal groups gives rise to undesirable metabolites for the development of sensory characteristics.

The best scores were obtained on farms F2 and F7 (with 36 h fermentation), which showed complete fermentation. In farm F2, in the metataxonomic observation of bacteria, the high abundance of *Leucnostoc* and *Weisella* stands out, and in farm F7, the abundance of the “Others” group and *Secundilactobacillus* stands out. Although there is also some presence of bacteria of the genus *Acetobacter* and *Bacillus* spp., unclassified Enterobacteria and yeasts in F2, the family *Saccharomycodaceae*-unclassified predominated, while in F7, f_*Saccharomycetaceae*_unclassified predominated.

On the other hand, in five farms, the fermentation was incomplete, so the quality of the coffee did not change or did not improve; therefore, it can be concluded that the time used for fermentation was not enough to guarantee the removal of the mucilage and improve its sensory quality. If the fermentation had reached its end, in at least 15 farms (75%), the wet fermentation process could have increased the final quality of the coffee. The use of the Fermaestro^®^ tool, which indicates the final fermentation time, could have significantly improved the process on all farms. With these results, it is possible to show the potential of the region in the production of special coffee (scores > 80) SCA, with higher coffee attributes related to acidity, fragrance and aroma, and different coffee sensory descriptors associated with the longest fermentation times.

## 5. Conclusions

The natural microbial diversity that the Cesar region possesses is involved in the processes of spontaneous fermentation in coffee. Although the microbial diversity in coffee fermentation is specific to the geographical region, the specific conditions of each farm that combine altitude, temperature, and water quality, among others, could determine the proportions and interactions that occur between microbial groups and, therefore, the production of certain metabolites that favor (or do not) the sensory characteristics of coffee in a cup. The metataxonomic analyses and the isolated microorganism suggest that the central microbiome in coffee fermentation from Cesar is composed mainly of *Enterobacter*, LAB of the genera *Leuconostoc* and *Lactiplantibacillus plantarum*, AAB with *Gluconobacter* and yeasts of the genus *Hanseniaspora*, which are associated with the region. The differences between the microorganism genera could be influenced by the coffee variety, as has been shown in other investigations; this is because although most of the results correspond to the farms with Var. Colombia and Var. Castillo General, the only farm with Var. Cenicafé 1 showed differences between bacterial and fungal genera that need to be better understood. Finally, the farms with the longest fermentation time (36 to 42 h) showed the highest final total acidity measure in the mucilage and highest coffee cup sensory attributes, as follows: acidity, fragrance and aroma, and special descriptors such as blackberries and fruity flavors.

## Figures and Tables

**Figure 1 foods-13-00839-f001:**
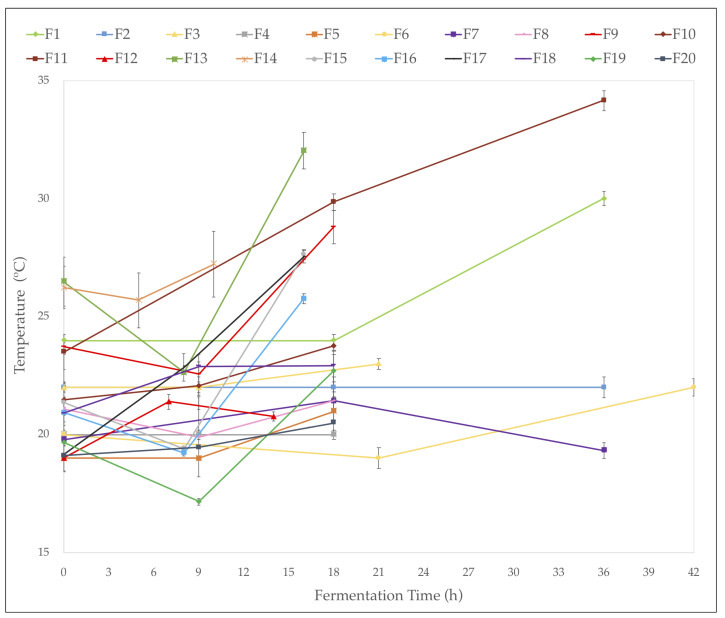
Temperature in the fermentation tanks during the fermentation process in the 20 farms identified: F1 to F20 in the department of Cesar.

**Figure 2 foods-13-00839-f002:**
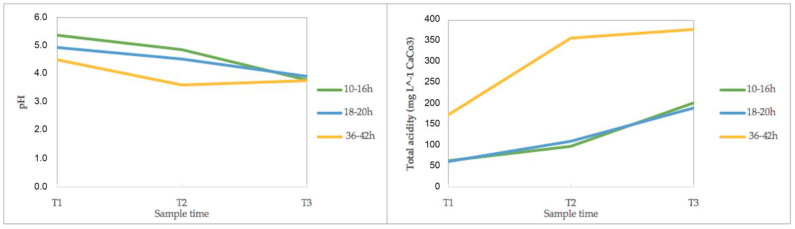
Physicochemical results of pH and total acidity of the mucilage during coffee fermentation in 20 farms. The data were grouped into three groups according to the farm’s final fermentation time. T1 (zero hours of fermentation). T2 (middle of the fermentation time). T3 (end of the fermentation time).

**Figure 3 foods-13-00839-f003:**
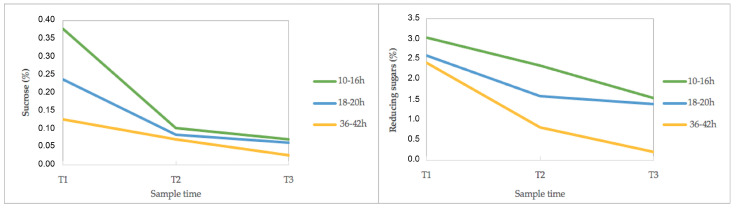
Physicochemical results of sucrose and reducing sugar of the mucilage during coffee fermentation in the 20 farms. The data were grouped into three groups according to the farm’s final fermentation time. T1 (zero hours of fermentation). T2 (middle of the fermentation time). T3 (end of the fermentation time).

**Figure 4 foods-13-00839-f004:**
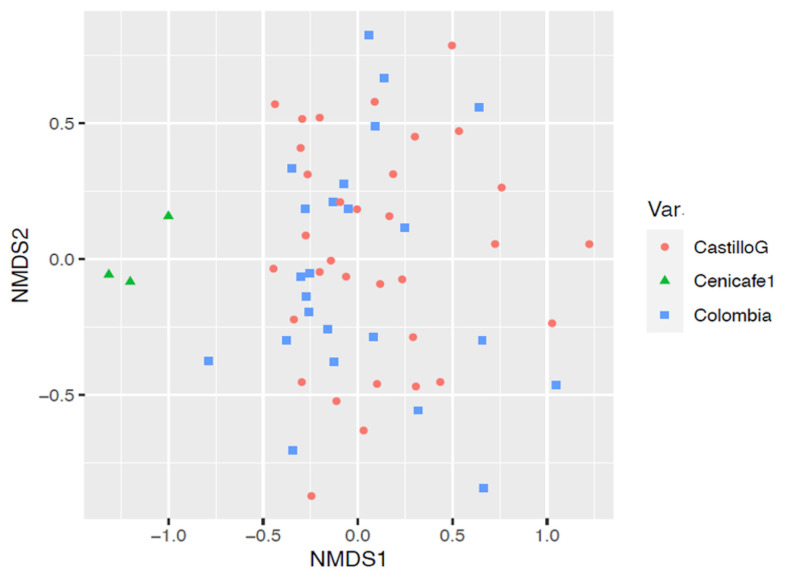
Nonmetric multidimensional scaling of the bacterial genus in mucilage from coffee samples during the fermentation process.

**Figure 5 foods-13-00839-f005:**
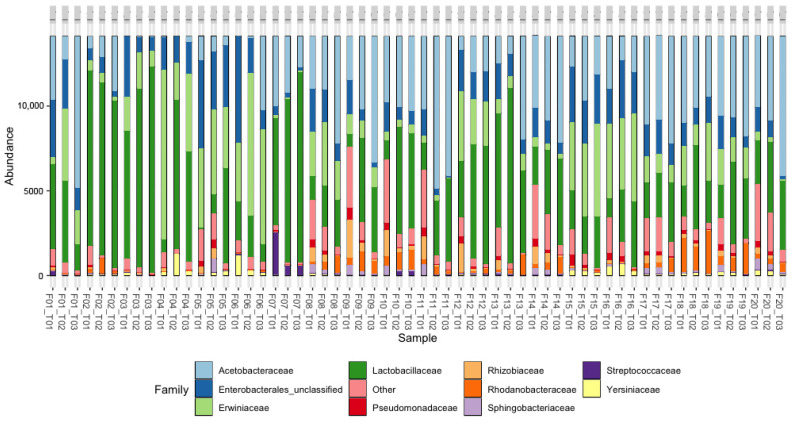
The relative abundance of sequences (%) corresponding to the bacterial taxonomic assignment to family category in coffee fermentation processes in 20 farms (Fs) in the department of Cesar. T01: Start of fermentation. T02: Midpoint during the fermentation and T03: end of fermentation.

**Figure 6 foods-13-00839-f006:**
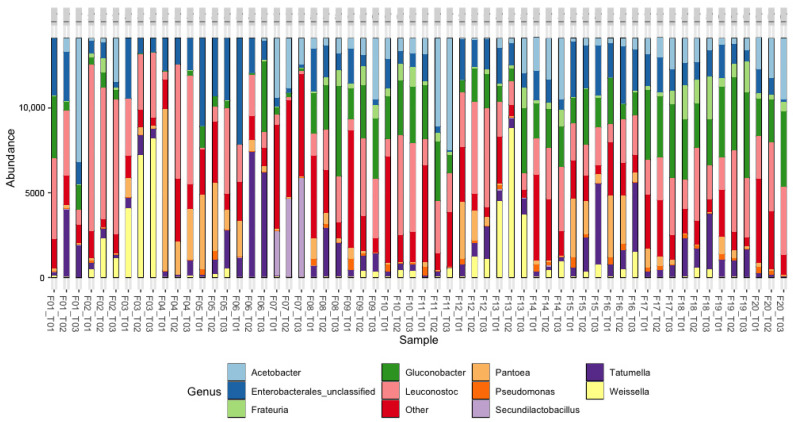
The relative abundance of sequences (%) corresponding to the bacterial taxonomic assignment to the genus category in coffee fermentation processes in 20 farms (Fs) in the department of Cesar. T01: Start of fermentation. T02: Midpoint during the fermentation and T03: end of fermentation. Detected OTUs below 0.5% are indicated as “Others”.

**Figure 7 foods-13-00839-f007:**
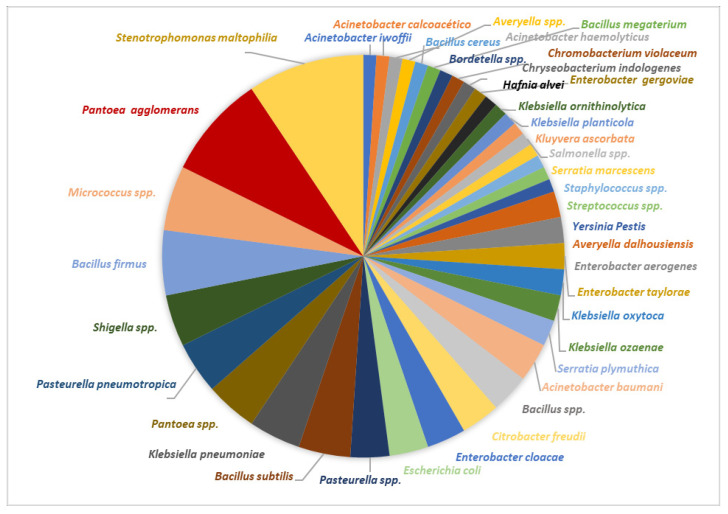
Mesophilic aerobic bacteria isolated during the coffee fermentation process at three different times in the 20 studied farms.

**Figure 8 foods-13-00839-f008:**
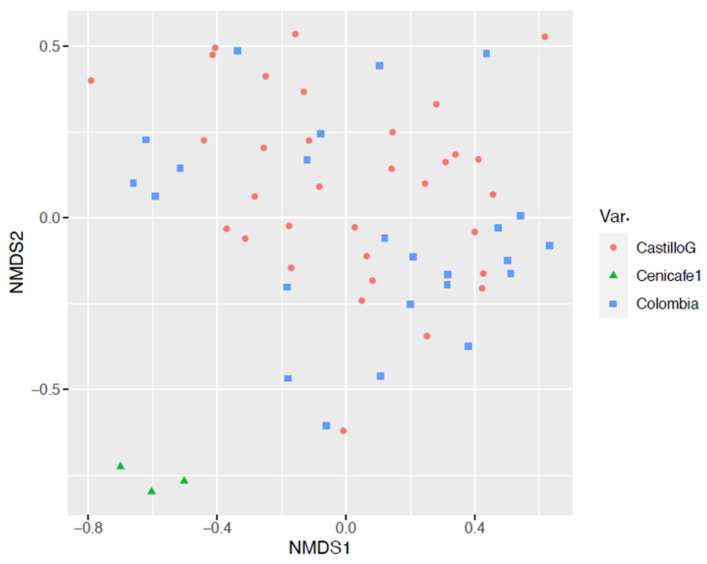
Nonmetric multidimensional scaling of fungi genera in mucilage from coffee samples during the fermentation process.

**Figure 9 foods-13-00839-f009:**
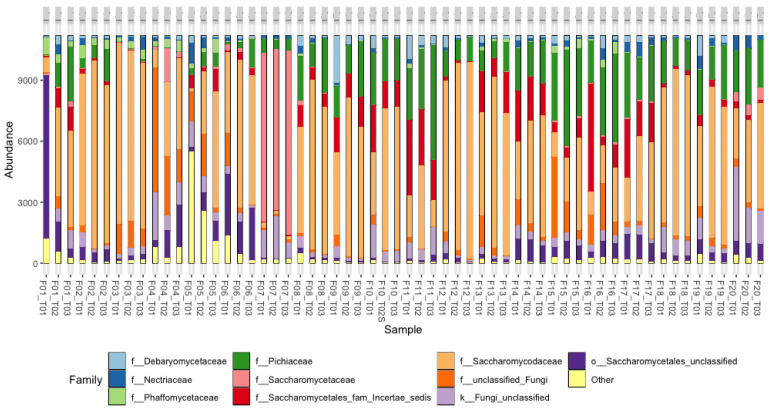
The relative abundance of sequences (%) corresponding to the fungi taxonomic assignment to the family level in coffee fermentation processes in 20 farms (Fs) in the department of Cesar. T01: Start of fermentation. T02: Midpoint during the fermentation and T03: end of fermentation. Detected OTUs below 0.5% are indicated as “Others”.

**Figure 10 foods-13-00839-f010:**
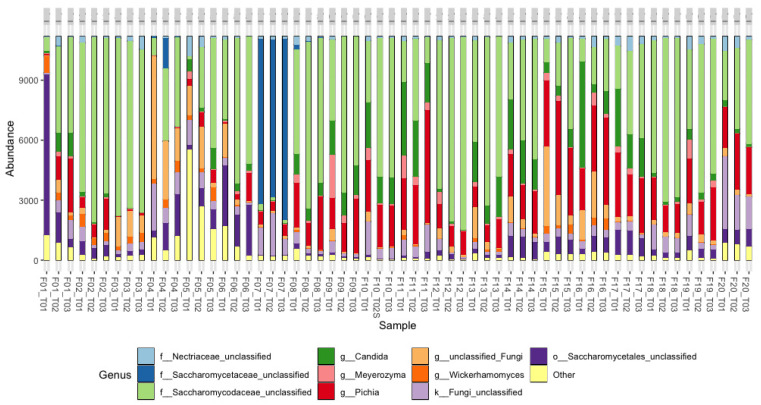
Relative abundance of sequences (%) corresponding to the fungi taxonomic assignment to the genus level in coffee fermentation processes in 20 farms (Fs) in the department of Cesar. T01: Start of fermentation. T02: Midpoint during the fermentation and T03: end of fermentation. Detected OTUs below 0.5% are indicated as “Others”.

**Figure 11 foods-13-00839-f011:**
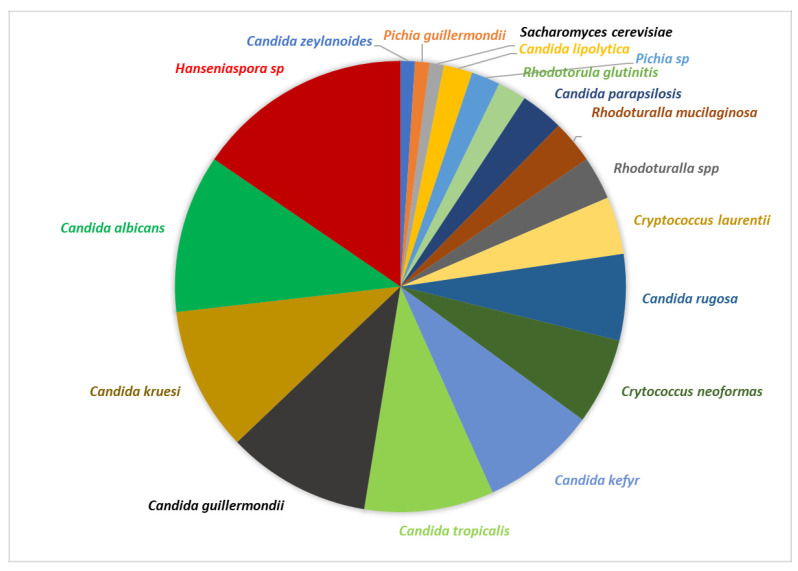
Yeast isolates during the coffee fermentation process at three different times in the 20 studied farms.

**Figure 12 foods-13-00839-f012:**
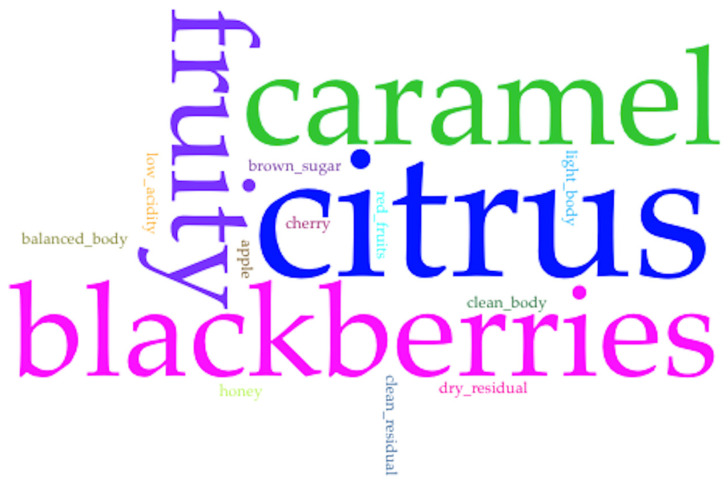
Word Cloud that shows the predominance of sensory descriptors in the coffee cup of farms with fermentation of 36–42 h.

**Table 1 foods-13-00839-t001:** Microbial counts during the fermentation process in 20 farms in the department of Cesar. Data are grouped into three groups according to the farm’s final fermentation time.

Fermentation Time (h)/Farms (Fs)	Sample Time	Mesophiles			Lactic Acid Bacteria			Coliforms			Yeasts			Filamentous Fungi		
		(Log10 CFUs/mL)														
		X		SD	X		SD	X		SD	X		SD	X		SD
10–16 h (F14 − F12 − F13 − F15 − F16 − F17)	T1	5.48	±	0.11	6.57	±	0.65	4.62	±	0.45	5.75	±	0.12	2.40	±	1.86
	T2	5.44	±	0.19	6.52	±	0.65	3.56	±	1.76	5.68	±	0.14	2.78	±	1.42
	T3	5.35	±	0.40	6.44	±	0.60	3.80	±	1.87	5.73	±	0.34	2.30	±	1.79
18–20 h (F3 − F4 − F5 − F8 − F9 − F10 − F18 − F19 − F20)	T1	5.50	±	0.302	6.61	±	0.55	4.22	±	0.85	5.68	±	0.26	3.42	±	0.34
	T2	5.49	±	0.436	6.65	±	0.58	4.13	±	0.79	5.55	±	0.36	3.58	±	0.21
	T3	5.62	±	0.413	6.27	±	1.28	3.33	±	2.04	5.65	±	0.33	3.04	±	1.20
36–42 h (F1 − F2 − F6 − F7 − F11)	T1	5.57	±	0.202	5.80	±	2.35	3.99	±	0.85	5.75	±	0.21	2.93	±	1.70
	T2	5.67	±	0.482	5.18	±	2.42	3.53	±	0.64	5.85	±	0.26	2.94	±	1.70
	T3	5.76	±	0.539	4.98	±	2.45	3.66	±	0.93	5.82	±	0.40	2.83	±	1.69

T1 (zero hours of fermentation). T2 (middle of the fermentation time). T3 (end of the fermentation time).

**Table 2 foods-13-00839-t002:** Identification and frequency of lactic acid bacteria (LAB) isolated into pure culture in coffee mucilage during the fermentation process at three different times in the 20 studied farms.

Microorganism	Time	Farms																				Frequency by Farms	
		F1	F2	F3	F4	F5	F6	F7	F8	F9	F10	F11	F12	F13	F14	F15	F16	F17	F18	F19	F20	#	%
*Fructilactobacillus fructivorans*	T1																					1	5.0
	T2														x								
	T3																						
*Lactiplantibacillus plantarum*	T1				x	x	x	x	x	x	x	x	x	x	x	x	x	x		x		17	85.0
	T2					x	x	x	x	x	x	x		x	x	x	x	x	x	x	x		
	T3							x	x		x	x		x	x	x		x		x	x		
*Lactiplantibacillus pentosus*	T1				x						x										x	6	30.0
	T2				x						x												
	T3	x											x							x			
*Lactobacillus delbrueckii*	T1	x																				2	10.0
	T2		x																				
	T3																						
*Lactococcus raffinolactis*	T1																		x			1	5.0
	T2																						
	T3																						
*Leuconostoc mesenteroides*	T1		x	x									x	x		x				x		9	45.0
	T2			x										x	x	x							
	T3		x	x	x	x								x		x							
*Leuconostoc citreum*	T1															x	x					3	15.0
	T2			x													x						
	T3																						
*Levilactobacillus brevis*	T1							x				x										8	40.0
	T2											x	x							x			
	T3									x				x					x		x		

T1 (zero hours of fermentation). T2 (middle of the fermentation time). T3 (end of the fermentation time).

**Table 3 foods-13-00839-t003:** Identification and frequency of AAB isolated into pure culture in coffee mucilage during the fermentation process at three different times in the 20 studied farms.

Microorganism	Time	Farms																				Frequency by Farms	
		F1	F2	F3	F4	F5	F6	F7	F8	F9	F10	F11	F12	F13	F14	F15	F16	F17	F18	F19	F20	#	%
*Acetobacter* spp.	T1											x					x			x		5	25.0
	T2																			x			
	T3											x				x			x	x			
*Acidomonas* spp.	T1																					1	5.0
	T2									x													
	T3																						
*Gluconobacter* spp.	T1							x						x								5	25.0
	T2									x							x		x				
	T3																						
*Sacharibacter* spp.	T1																		x			2	10.0
	T2																						
	T3																	x					

T1 (zero hours of fermentation). T2 (middle of the fermentation time). T3 (end of the fermentation time).

**Table 4 foods-13-00839-t004:** Total SCA score in the coffee cup in 20 farms in the department of Cesar. Data were grouped into three groups according to the farm’s final fermentation time.

Fermentation Time (h)/Farms (Fs)				Sample Time					
	T1			T2			T3		
	X		SD	X		SD	X		SD
10–16 h (F12 − F13 − F14 − F15 − F16 − F17)	82.7	±	0.86	82.7	±	0.43	82.5	±	0.86
18–20 h (F3 − F4 − F5 − F8 − F9 − F10 − F18 − F19 − F20)	82.3	±	1.43	82.8	±	6.96	82.5	±	0.71
36–42 h (F1 − F2 − F6 − F7 − F11)	82.5	±	1.60	82.7	±	8.90	83.7	±	0.54

SCA score: 90–100 outstanding, 85.0–89.99 excellent, 80–84.99 very Good, <80 commercial coffee.

**Table 5 foods-13-00839-t005:** Score of some of the sensory attributes identified in the coffee cup in 20 farms in the department of Cesar. Data were grouped into three groups according to the farm’s final fermentation time.

Fermentation Time (h)/Farms (Fs)	Sample Time	Fragance/Aroma			Flavor			Acidity		
		X		SD	X		SD	X		SD
10–16 h (F12 − F13 − F14 − F15 − F16 − F17)	T1	7.55	±	0.15	7.61	±	0.14	7.55	±	0.10
	T2	7.65	±	0.09	7.54	±	0.13	7.54	±	0.07
	T3	7.50	±	0.11	7.57	±	0.13	7.52	±	0.14
18–20 h (F3 − F4 − F5 − F8 − F9 − F10 − F18 − F19 − F20)	T1	7.54	±	0.22	7.53	±	0.24	7.52	±	0.25
	T2	7.53	±	0.09	7.56	±	0.17	7.56	±	0.13
	T3	7.55	±	0.09	7.54	±	0.12	7.50	±	0.11
36–42 h (F1 − F2 − F6 − F7 − F11)	T1	7.65	±	0.27	7.58	±	0.17	7.55	±	0.19
	T2	7.69	±	0.16	7.54	±	0.19	7.60	±	0.19
	T3	7.69	±	0.13	7.85	±	0.12	7.73	±	0.21

## Data Availability

The datasets presented in this study can be found in online repositories. The names of the repositories and accession number(s) can be found at https://www.ncbi.nlm.nih.gov/ (accessed on 30 May 2023) Bioproject: PRJNA977378 and PRJNA977712.

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
