# Peer review of "Metataxonomic Identification of Microorganisms during the Coffee Fermentation Process in Colombian Farms (Cesar Department)"

_foods, 2024, doi:10.3390/foods13060839_

Round 1

Reviewer 1 Report

Comments and Suggestions for Authors

The work is relevant and the data generated has great potential to bring important insights to the area of fermentation microbiology as well as the coffee production chain.
However, many adjustments must be made both in writing and in data analysis.

All recommendations for changes must be met. Suggestions are for improvement.

Therefore, below are the checkpoints to be corrected by the authors:

INTRODUCTION SECTION IMPROVEMENTS

 Lanes 39-41: Please insert references that validate these statements.

 Lanes 42-45: I consider reference "#2" to be supporting this sentence. I recommend inserting it at the end of this sentence. Leaving the reference "#1" just to support the next sentence that deals with the consumer market.

 Lanes 59-63: I believe that the authors are not contradicting the previous paragraph here. Therefore, the expression "on the other hand" is not appropriate.

 Lane 85: I believe that the authors are not contradicting the previous paragraph here. Therefore, the expression "on the other hand" is not appropriate.

 Lane 104: replace "where" with "in which"

 Lanes 116-117: Their methods do not allow identifying the microbiota responsible for the fermentation process. To achieve this objective, isolation of microorganisms and fermentative potential tests should have been carried out. I believe that the best that their methods allow is to characterize the microbiota present in the microenvironments linked to the coffee fermentation process on the farms that make up the proposed sampling design. I strongly recommend adapting the main goal.

 Lanes 119-123: There is an inconsistency here:

You have established here that the objective of this manuscript is to identify the microbiota responsible for the fermentation process. However, several other analyzes relating to the fermentation process (mostly relating to process quality control) were carried out. To achieve microbiota identification, none of the process quality control analyzes are necessary. Therefore, I believe it is necessary in this paragraph to describe the purpose of carrying out such quality control analyzes and how they are linked to the main goal (I believe that at this point you will realize that the main objective is broader than what is presented).

 METHODS SECTION IMPROVEMENTS

 Lanes 232-234: There is a conception mistake here:

What was performed here was the use of amplicon sequencing with the purpose of carrying out a meta-taxonomic approach to characterize the taxonomic diversity of the microenvironments or microbiomes accessed through sampling. Molecular markers are just tools to estimate how many and which groups there are. Diversity (alpha, beta and gamma) is estimated using statistical methods (which were mentioned later in the manuscript).

 Lane 244: What is "deep" sequencing? The correct term is "high throughput sequencing"

 Lane 261: OTU

 Lanes 261-266: Permanova is applicable for multivariate analyzes which, specifically for your data, only apply to the estimated beta diversity matrices.
The alpha-diversity vectors generated by the estimators used (Chao1, ACE, Shannon, ...) must have been analyzed using mean tests (probably Kruskal-Wallis).
ACE is only recommended when obtaining a much larger number of samples than those collected by you. I recommend not using this estimator.
I strongly recommend that you rewrite this paragraph by making adjustments to the data analysis description in the correct way.

 RESULTS SECTION IMPROVEMENTS

 Here the authors identified key characteristics in each of the farms accessed (location, fermentation process, tank material, and others).
However, the authors chose to present the results by farm. I believe that this approach does not provide useful insights, both academically and for the production chain. Presenting individual results is only of interest to individual coffee producers.
I suggest that the results be presented by grouping the farms by their key attributes and comparing the variability within and between groups. I believe that in this way patterns can emerge and insights can be identified from associations and correlations between groups and evaluated parameters.

 Lane 401: Please delete ":"

 DISCUSSION SECTION IMPROVEMENTS

 The manuscript lacks a final synthesis, including a discussion on the impacts of the results on the practice of coffee fermentation.
Furthermore, the work presents a plethora of limitations due to the chosen (or available) methods and no discussion of this was presented in the discussion.

Lanes 923-929: Please indicate here which analysis supports this statement. I did not find tests in the results section that support this statement since the data were presented individually.

 Lanes 967-971: Carrying out analyzes by groups could generate potential associations between common attributes between farms and the analyzed parameter.

I reinforce the need to carry out analyzes by groups.

 Lanes 1028-1033: Please indicate here which analysis supports this statement.

 CONCLUSION SECTION 

 Finally, and crucially important, the manuscript lacks a conclusion section.
It is essential that these points are added to the text.

 TABLE IMPROVEMENTS

 Table 1: I suggest that this table could be presented as supplementary material.

 Table 2: What patterns can we verify when the results are presented individually?

I strongly recommend reorganizing the data into groups formed based on the attributes presented in Table 1.

 Table 3: Similar recommendation to Table 2.

 Table 6. Similar recommendation to Table 2.

 FIGURE IMPROVEMENTS

 Figure 2: Was this analysis performed at the gender level? In the methodology section it was described that this analysis was carried out based on the OTU table.

 Figure 6: Was this analysis performed at the genus level? In the methodology section it was described that this analysis was carried out based on the OTU table.

 Figure 8: I do not see the meaning of presenting these data at the genus level since the OTUs with the highest abundance were not identified at the genus level.
I suggest only presenting the taxa bar plot at the family level.
Comments on the Quality of English Language

Extensive editing of English language required

Author Response

Foods-284642-R1

 Dear Reviewers

The authors would like to thank the reviewer N°1-2 for all the comments that helped us to improve the quality and clarity of the manuscript.  In the new version of the paper:

- All the information that we want to take out from the manuscript are crossed out.

-The specific changes suggested by the reviewer are highlight in yellow color

-Changes with respect to the original documents are in red and high light in yellow color

-New information added to the document is in red color.

-Changes in order to avoid repetition rate are highlight in Green

In order to improve the English grammar the document was sent for professional MDPI English Editing Service.

Reviewer 1

Comments and Suggestions for Authors

The work is relevant and the data generated has great potential to bring important insights to the area of fermentation microbiology as well as the coffee production chain.
However, many adjustments must be made both in writing and in data analysis.

All recommendations for changes must be met. Suggestions are for improvement.

Therefore, below are the checkpoints to be corrected by the authors:

Authors Comments.

We appreciate very much the reviewer’s suggestion for improving the document. We tried to make all the possible correction and the advice of grouping part of the information, modified some of the table and put some table in the supplementary material was very appropriated because it helps to make clearer the whole manuscript and the work in general. Thank you for your input.

Also, we think that the suggestion about the change of the objective to make it more wide help also to understand in a better way the work.

INTRODUCTION SECTION IMPROVEMENTS

  1. Lanes 39-41: Please insert references that validate these statements.

Authors Comments

The reference is the same one to the follow sentence (reference 1), and it was introduced in the document line 44.

  1. Lanes 42-45: I consider reference "#2" to be supporting this sentence. I recommend inserting it at the end of this sentence. Leaving the reference "#1" just to support the next sentence that deals with the consumer market.

                Authors Comments

The change was done as the reviewer advice -reference 1 is in line 49 and- reference 2 is in line 51.

  1. Lanes 59-63: I believe that the authors are not contradicting the previous paragraph here. Therefore, the expression "on the other hand" is not appropriate.

Lane 85: I believe that the authors are not contradicting the previous paragraph here. Therefore, the expression "on the other hand" is not appropriate.

 Authors Comments

Reviewer is right.  Line 88- On the other hand "was change for Additionally.

  1. Lane 104: replace "where" with "in which"

             Authors Comments

The change was done as the reviewer advice in line 106.

  1. Lanes 116-117: Their methods do not allow identifying the microbiota responsible for the fermentation process. To achieve this objective, isolation of microorganisms and fermentative potential tests should have been carried out. I believe that the best that their methods allow is to characterize the microbiota present in the microenvironments linked to the coffee fermentation process on the farms that make up the proposed sampling design. I strongly recommend adapting the main goal.

            Authors Comments

Reviewer is right. Because of the comments, we change the objective according to the methodology and it really show a wider scope of the work. The changes are in lines 119 to 123.

The objective of this research was to characterize the microbiota present during the coffee fermentation processes in 20 farms in the department of Cesar with the usual postharvest practices in each farm and to determine the relations among micro-orrganisms, quality control of the fermentation process, farm location and coffee quality.

  1. Lanes 119-123: There is an inconsistency here:

You have established here that the objective of this manuscript is to identify the microbiota responsible for the fermentation process. However, several other analyzes relating to the fermentation process (mostly relating to process quality control) were carried out. To achieve microbiota identification, none of the process quality control analyzes are necessary. Therefore, I believe it is necessary in this paragraph to describe the purpose of carrying out such quality control analyzes and how they are linked to the main goal (I believe that at this point you will realize that the main objective is broader than what is presented).

Authors Comments

Reviewer is right and base on this observation we change the objective to include the 4 aspects that we really study during the process of fermentation in this work and that are linked together: Microorganisms, Quality control of the fermentation process, Farm Location and Coffee Quality. The changes were done in lines 121 to 123.

 METHODS SECTION IMPROVEMENTS

  1. Lanes 232-234: There is a conception mistake here:

What was performed here was the use of amplicon sequencing with the purpose of carrying out a meta-taxonomic approach to characterize the taxonomic diversity of the microenvironments or microbiomes accessed through sampling. Molecular markers are just tools to estimate how many and which groups there are. Diversity (alpha, beta and gamma) is estimated using statistical methods (which were mentioned later in the manuscript).

Authors Comments

Reviewer is right and base on that observation we change the document in lines 232 and 233.

(For the samples of DNA, amplicons were generated and they were used to characterize the microbiome taxonomic diversity)

Also, in the abstract in line 18 we included the information also in line 18.

  1. Lane 244: What is "deep" sequencing? The correct term is "high throughput sequencing"

 Authors Comments

Reviewer is right. The term Deep sequencing experiment (NGS) in Lines 231 and 245 were change to the term High throughput sequencing. Also, in the supplementary material in Table S5 and S7 the change was done.

  1. Lane 261: OUT

Authors Comments

The change was done in line 262.

  1. Lanes 261-266: Permanova is applicable for multivariate analyzes which, specifically for your data, only apply to the estimated beta diversity matrices.
    The alpha-diversity vectors generated by the estimators used (Chao1, ACE, Shannon, ...) must have been analyzed using mean tests (probably Kruskal-Wallis).
    ACE is only recommended when obtaining a much larger number of samples than those collected by you. I recommend not using this estimator.
    I strongly recommend that you rewrite this paragraph by making adjustments to the data analysis description in the correct way.

Authors Comments

Reviewer is right. We re-write the paragraph from line 262-271 and delete the wrong information. 

RESULTS SECTION IMPROVEMENTS

  1. Here the authors identified key characteristics in each of the farms accessed (location, fermentation process, tank material, and others).
    However, the authors chose to present the results by farm. I believe that this approach does not provide useful insights, both academically and for the production chain. Presenting individual results is only of interest to individual coffee producers.
    I suggest that the results be presented by grouping the farms by their key attributes and comparing the variability within and between groups. I believe that in this way patterns can emerge and insights can be identified from associations and correlations between groups and evaluated parameters.

Authors Comments

Base on the reviewer´s comments, Physico-chemical data, micro-organisms counts, Total             SCA score and Attribute Analysis from all 20 farms were grouped into three groups       according to the final fermentation time. The first group contains the farms with up to 16 h     of fermentation. The group 2 farms had 18 to 20 h of fermentation, and the group 3 farms   had a fermentation time of between 36 and 40 h. The data average and standard deviation                 were calculated in order to see the differences among the groups.

This kind of grouping allow us to have a better understanding of the different processes and had a better discussion. New correlations were identified. The longest fermentation time in the farms a higher total acidity and higher score in some cup attributes were identifies.

The table with all the physicochemical data was send to the supplementary material and two New figures (Figure 2-line 474 and Figure 3-line 521) with data corresponding to pH, Total acidity, sugar contents were added. The data of Brix were not presents as figure because the behavior was very similar to the one obtained with the pH results.

Also, the data related to the microorganisms’ counts was replaced for a new table, table 1 in line 566. The original table with all the data was leaved as supplementary material.

In the same way a new table (Table 4) with the Sensory quality and SCA score was added in line 929, and another with the sensory attribute was added in line 961. The information with all data from the farms was included as attached document.

  1. Lane 401: Please delete ":"

Authors Comments

            It was deleted in the subtitle 3.2 Physicochemical analysis, line 426.

DISCUSSION SECTION IMPROVEMENTS

  1. The manuscript lacks a final synthesis, including a discussion on the impacts of the results on the practice of coffee fermentation.

            Authors Comments

In the discussion was added information of the impact of the good fermentation practices during the coffee fermentation. This was included Line 1005, also in Line 1195-1197. Also, a section with conclusion was included.

  1. Furthermore, the work presents a plethora of limitations due to the chosen (or available) methods and no discussion of this was presented in the discussion.

            Authors Comments

Base on the new way of presenting some of the results the discussion is easier to understand. The grouping of the data and classification of the farms according the fermentation time allow to follow better the discussion.  New discussion about methods and results that were not discussed previously were added. For example, goods practices and the effect of the long fermentation in the attributes of the coffee cup and use of Fermaestro.

  1. Lanes 923-929: Please indicate here which analysis supports this statement. I did not find tests in the results section that support this statement since the data were presented individually.

Authors Comments

This information is now in the line 1081 to 1087- some changes were done in the paragraph. In this case the new Table 1. Microbial counts during the fermentation process in 20 farms in the department of Cesar. Data grouped into 3 groups according to the farm final fermentation time, allows to see clearly how the microorganism population behave depending of the fermentation time.

  1. Lanes 967-971: Carrying out analyzes by groups could generate potential associations between common attributes between farms and the analyzed parameter.

I reinforce the need to carry out analyzes by groups.

Authors Comments.

                The advice was taking and manuscript reorganized. The particular information from lines            1119 and 1123 was deleted.

  1. Lanes 1028-1033: Please indicate here which analysis supports this statement.

Authors comments

In the new document the information is in line 1181 to 1187. The new analysis supports the statement- Table 1. The paragraph has small changes.

 CONCLUSION SECTION 

  1. Finally, and crucially important, the manuscript lacks a conclusion section.
    It is essential that these points are added to the text.

Authors comments

A conclusion section was included at the endo of the document from line 1223 -1249

 TABLE IMPROVEMENTS

 19.Table 1: I suggest that this table could be presented as supplementary material.

Authors comments

The original Table 1 with information about all the farms was moved to the Supplementary Material.

  1. Table 2: What patterns can we verify when the results are presented individually?

Authors comments

The original Table 2 with all the physicochemical information was moved to supplementary material. Two new set of figures (Figure 1 and 2)   with data grouped into 3 group according to fermentation time, as it was described previously in point 11 of this document was added.

21.I      strongly recommend reorganizing the data into groups formed based on the attributes presented in Table 1.

Authors comments

Reviewer is right, the change was done.

  1. Table 3: Similar recommendation to Table 2.

Authors comments

The recommendation was taken. The original Table 3 was moved to supplementary material a new table- grouping the farm data by fermentation time was added. In the new document is Table 1. Line 566.

  1. Table 6. Similar recommendation to Table 2.

Authors comments

The recommendation was taken. The original Table 6 was moved to supplementary material a new table 4 grouping the Total SCA score of coffee cups was done- Line 929.

FIGURE IMPROVEMENTS

  1. Figure 2: Was this analysis performed at the gender level? In the methodology section it was described that this analysis was carried out based on the OTU table.

Figure 6: Was this analysis performed at the genus level? In the methodology section it was described that this analysis was carried out based on the OTU table.

Authors comments

For figures 2 and 6, the nonmetric multidimensional scaling (NMDS) was done at level of genus, some cases the information of species was also available. To make clear in the supplementary tables S5 and S6 where the OTUs are reported we add in column 7 the information Genus-species. The information is also included in lines 617-621.

  1. Figure 8: I do not see the meaning of presenting these data at the genus level since the OTUs with the highest abundance were not identified at the genus level.
    I suggest only presenting the taxa bar plot at the family level.

Authors comments

With the OTUS and using the Mothur analysis a taxonomic assignment was generated and the taxa level goes up to genus and sometime species.

In our case the knowledge of the genus is important and we would like to keep this information in the document. Additionally, we know that the microorganism’s genus levels identified are right because, in addition we performed shotgun metagenomic sequencing in some of the samples to confirm the most abundant genus and species in the samples. We have that information but it was not included in the document. We hope to be able to published it in other paper.

Reviewer 2 Report

Comments and Suggestions for Authors

The authors present a very comprehensive study of microbioma in an extremely long paper. This is a very relevant and interesting topic, however the lenght of the paper makes the results and significance of findings hard to grasp.

I suggest the authors review the discussion section and in particular add some discussion on the relevance and the meaning of the results, not only discussing quantiative results, which are already extensively covered in the results section.

The paragraph L1034-L1039 is what it probably should be the most exciting finding of this paper, however the information in this paragraph doesn't tell much, it is only a summary of the results, not a discussion. This is a very important aspect of the paper to link fermentation to sensory results and a more careful assesment of the results and the relevance should be discussed in this paragraph.

L1070 - Could the authors discuss potential approaches that would allow for interaction of fermentation and resulting flavour to be better assessed? Can the authors speculate if an analysis of their data could provide more information? Where is the missing part that the authors didn't do this? is the missing part the robustness of the sensory data or the lack of relation?

Comments on the Quality of English Language

The presentation of tables and graphs (labels, units) as well as formating should be improved.

For example:

Table 2 - column captions formating. Units for total acidity

Table 6 - column captions formatiing

Author Response

Foods-284642-R1

 Dear Reviewers

The authors would like to thank the reviewer 1 and 2 for all the comments that helped us to improve the quality and clarity of the manuscript.  In the new version of the paper:

- All the information that we want to take out from the manuscript are crossed out.

-The specific changes suggested by the reviewers are highlight in yellow color

-Changes with respect to the original documents are in red color and high light in yellow color

-New information added to the manuscript is in red color.

-Changes in order to avoid repetition rate are highlight in Green

In order to improve the English grammar, the document the document was sent for professional MDPI English Editing Service.

Reviewer 2

  1. The authors present a very comprehensive study of microbioma in an extremely long paper. This is a very relevant and interesting topic, however the lenght of the paper makes the results and significance of findings hard to grasp.

Autor Comments

We appreciate very much the reviewer’s suggestion for improving the document. We tried to make all the possible correction. First, we intended to reduce the length of the manuscript for this: two large tables with information about each one of the farms were included in the supplementary material making the document easier to read.

We delete a table with information about the farms with incomplete fermentation that was not necessary to understand the results. In the new document information not relevant was deleted and it is seen crossed out.

  1. I suggest the authors review the discussion section and in particular add some discussion on the relevance and the meaning of the results, not only discussing quantiative results, which are already extensively covered in the results section.

Autor Comments

Base on the reviewer´s comments, Physico-chemical data, micro-organisms counts, Total SCA   score and Attribute Analysis from all 20 farms were grouped into three groups           according to the final fermentation time. The first group contains the farms with up to 16 h of fermentation. The group 2 farms had 18 to 20 h of fermentation, and the group 3 farms had a fermentation time of between 36 and 40 h. The data average and standard deviation were calculated in order to see the differences among the groups.

This kind of grouping allow us to have a better understanding of the different processes and had a better discussion. New correlations were identified. The longest fermentation time in the farms was correlated with a higher total acidity and higher score in some cup attributes were identifies.

We focus the discussion base on the new analysis and good fermentation practices.

Also, we deleted specific information of particular farms lines in the results and discussion.

  1. The paragraph L1034-L1039 is what it probably should be the most exciting finding of this paper, however the information in this paragraph doesn't tell much, it is only a summary of the results, not a discussion. This is a very important aspect of the paper to link fermentation to sensory results and a more careful assesment of the results and the relevance should be discussed in this paragraph.

Autor Comments

In the new document it corresponds to lines 1188 to 1197. We add more information in this paragraph indicating the importance of goods practices. Also, base on the advice of the reviewer, the coffee quality results were presented in a different way new tables 4 Line 929 and Table 5 Line 961, and we analyzed also the descriptors associated to those attributes, a new Figure 12. Word Cloud that shows the predominance of sensory descriptors in the coffee cup of farms with fermentation of 36-42h at time 3 was included because it showed the largest differences. Line 968.

  1. L1070 - Could the authors discuss potential approaches that would allow for interaction of fermentation and resulting flavour to be better assessed? Can the authors speculate if an analysis of their data could provide more information? Where is the missing part that the authors didn't do this? is the missing part the robustness of the sensory data or the lack of relation?

Autor Comments

Bases on the advice of the reviewer and  the new analysis of the results, we could identified new association. For example

The group of farms with the longest fermentation time independent of the coffee variety was also the one with the final highest total acidity measure in the mucilage and a high coffee cup SCA score at the end of the fermentation process. This score was related with the sensory attributes: acidity, fragrance and aroma and special descriptors such as blackberries and fruity flavors.

This information is in the discussion in line 1037-1041 and also was included in the abstract in line 32-33

  1. Comments on the Quality of English Language

Authors comments

Regarding the English revision. In order to check the English the document was send to MDPI for a high quality paid editing service.

  1. The presentation of tables and graphs (labels, units) as well as formating should be improved.

For example: Table 2 - column captions formating. Units for total acidity

Autor Comments

Acidity units was added to the figure 1

Table 6 - column captions formatiing.

Autor Comments

Table 6 was moved to supplementary material. Change was done

All the tables and figure were reviewed.
